# Structural modularity of the XIST ribonucleoprotein complex

Zhipeng Lu [1,5✉], Jimmy K. Guo[1], Yuning Wei[1], Diana R. Dou [1], Brian Zarnegar[2], Qing Ma[1,6], Rui Li[1], Yang Zhao[1], Fan Liu[1], Hani Choudhry[1,3], Paul A. Khavari [2] & Howard Y. Chang [1,2,4✉]

Long noncoding RNAs are thought to regulate gene expression by organizing protein complexes through unclear mechanisms. XIST controls the inactivation of an entire X chromosome in female placental mammals. Here we develop and integrate several orthogonal structure-interaction methods to demonstrate that XIST RNA-protein complex folds into an evolutionarily conserved modular architecture. Chimeric RNAs and clustered protein binding in fRIP and eCLIP experiments align with long-range RNA secondary structure, revealing discrete XIST domains that interact with distinct sets of effector proteins. CRISPR-Cas9-mediated permutation of the Xist A-repeat location shows that A-repeat serves as a nucleation center for multiple Xist-associated proteins and m$^6$A modification. Thus modular architecture plays an essential role, in addition to sequence motifs, in determining the specificity of RBP binding and m$^6$A modification. Together, this work builds a comprehensive structure-function model for the XIST RNA-protein complex, and suggests a general strategy for mechanistic studies of large ribonucleoprotein assemblies.

[1] Center for Personal Dynamic Regulomes, Stanford University, Stanford, CA 94305, USA. [2] Department of Dermatology, Stanford University School of Medicine, Stanford, CA 94305, USA. [3] Department of Biochemistry, Cancer Metabolism and Epigenetic Unit, Faculty of Science, Cancer and Mutagenesis Unit, King Fahd Center for Medical Research, King Abdulaziz University, Jeddah 22252, Saudi Arabia. [4] Howard Hughes Medical Institute, Stanford University, Stanford, CA 94305, USA. [5] Present address: Department of Pharmacology and Pharmaceutical Sciences, University of Southern California, 1985 Zonal Avenue, Los Angeles, CA 90089, USA. [6] Present address: Synthetic Biology Department, Shenzhen Institute of Advanced Technology, Chinese Academy of Sciences, 518055 Shenzhen, PR China. ✉email: zhipengl@usc.edu; howchang@stanford.edu

Long noncoding RNAs (lncRNAs) play essential roles in many aspects of gene expression in development and disease[1]. lncRNAs control X chromosome inactivation (XCI), genome imprinting, immune response, cell-cycle regulation, genome stability, lineage commitment, and embryonic stem cell (ESC) pluripotency[2–9]. The list of functional lncRNAs is growing rapidly as more studies are conducted in a wide variety of systems. lncRNAs are distinguished from mRNAs in their processing and ultimate mechanisms of action[10,11]. Accumulating evidence suggested that lncRNAs often serve as flexible scaffolds to recruit and coordinate multiple protein complexes to execute specific functions. For example, the yeast telomerase RNA recruits multiple proteins, and relocation of the protein-binding motifs does not disrupt the function of the telomerase complex[12]. The lncRNA HOTAIR recruits two distinct histone modification complexes, LSD1 and PRC2, to specify combinatorial patterns of histone modifications[13].

Several new experimental strategies have been developed and applied to lncRNAs to determine structures and interactions that underlie their functions, in particular Selective 2' Hydroxyl Acylation analyzed by Primer Extension (SHAPE) and dimethyl sulfate sequencing (DMS-seq) probe nucleotide (nt) accessibility, a proxy for RNA base pairing probability[14]. These two approaches have been applied to several in vitro transcribed lncRNAs, such as XIST, HOTAIR, COOLAIR, and Braveheart[15–18]. In vivo DMS-seq on the XIST RNA suggested functional local structure elements but did not reveal high-level organization[19]. These studies reported vaguely defined domains in these long transcripts, but it remains unclear whether these domains are relevant in physiological conditions. Computational modeling based on chemical probing is prone to errors, especially for long transcripts[20,21].

Female placental mammals have two X chromosomes while males only have one. The difference in gene dosage relative to autosomes is compensated by a mechanism called XCI, where one of the two X chromosomes in females is randomly silenced. XIST (X Inactive Specific Transcript) is an essential ~19 kb lncRNA that controls XCI by recruiting multiple proteins to deposit epigenetic modifications, remodel the X chromosome, and silence transcription in specific nuclear compartment[8,9]. Several studies used biochemical and genetic screens to find new players in XIST functions[22–26]. Xist associates with at least 81 proteins through direct RNA–protein and indirect protein–protein interactions. If occurring all together, the XIST ribonucleoprotein (RNP) is many times the size of the ribosome. A number of XIST-associated proteins mediate critical steps in XCI. For example, the RNA-binding protein (RBP) SPEN binds the A-repeat of XIST and recruits the SMRT-HDAC3 complex to repress transcription[24–27]. RBM15/RBM15B recruit the WTAP-METTL3-METTLE14 (WMM) RNA methyltransferase complex to install m6A, which are required for XIST function[28]. LBR recruits the XIST-coated Xi to nuclear lamina for efficient silencing[29], HNRNPU family proteins and CIZ1 attach the XIST RNA to the inactive X chromosome[30–34]. In particular, many of these studies have been performed in different systems (e.g., for CIZ1 and HNRNPU in chromatin tethering of XIST), such as somatic cells or mouse ESCs (mESCs), and the specific roles of these complexes in each biological context have not been resolved. A key question that remains is how these numerous proteins and the structured XIST RNA are assembled into a functional complex. Moreover, all of the above studies were done using mESCs, and how the human XIST RNP may be organized given substantial sequence divergence is not known.

In our prior work, we used three orthogonal methods, PARIS, icSHAPE, and conservation analysis, to demonstrate a modular architecture of the XIST RNA. We found that stochastic folding of the A-repeat domain serves as a multivalent platform to recruit SPEN. To determine the structural of the multi-functional XIST RNP complex, we develop and integrate several methods, including PARIS (psoralen crosslinking)[27], formaldehyde RNA immunoprecipitation sequencing (fRIP-seq; formaldehyde)[35], enhanced crosslinking and immunoprecipitation (eCLIP; ultraviolet (UV) crosslinking)[36], and PIRCh (purification of interacting RNA on chromatin, using glutaraldehyde crosslinking)[37]. Together, we find that the entire XIST RNA–protein complex is folded in a modular manner. XIST-associated proteins cluster together in the three-dimensional folded complex, instead of spreading along the linear sequence. The clustering of proteins on the XIST RNA structure predicts a modular organization of XIST functions. The folding of the RNA creates physical proximity that directs the m6A methylase and hence the modifications. Together, this analysis establishes a unifying model for XIST functions.

Using CRISPR-Cas9 genome editing, we reorganize the architecture of the XIST RNA by moving the A-repeat domain to other locations. This reorganization is followed by relocation of XIST-associated proteins, demonstrating a role of the architecture in separating the chromatin-binding and nuclear membrane-binding regions of the XIST RNP, and a role of the domain architecture in guiding protein binding to the RNA, and the m6A modification of the RNA, which is required for XIST functions. Together, this study builds a comprehensive model for the XIST RNP and establishes a paradigm for studying the structural basis of lncRNA functions.

## Results

**RNA chimeras in fRIP-seq reveal modular XIST RNP architecture.** Using PARIS, icSHAPE, and structure conservation, we demonstrated that the XIST RNA is folded into modular and compact domains, each spanning hundreds to thousands of nts[27]. While a handful of XIST-binding RBPs have been mapped to distinct regions on XIST, the overall organization of this RNP remains unknown. We reasoned that, in addition to psoralen, other chemical crosslinkers, together with proximity ligation, could capture the higher-order architecture of XIST, therefore providing additional lines of evidence for XIST structure and RNA–protein interactions.

We searched published RNA–protein crosslinking studies and found RNA–RNA chimeras for XIST in a set of fRIP-seq experiments targeting 24 RBP and chromatin-associated proteins in K562 cells, a female human myeloid cell line that undergoes XCI[35,38,39]. Briefly, cells were lightly crosslinked with formaldehyde to fix RNA–protein interactions, sonicated to small fragments, and then RNA–protein complexes were immunoprecipitated (Fig. 1a). The antibodies used in fRIP-seq have been extensively validated as part of the ENCODE project. However, it is important to recognize the caveat that there may still be minor cross-reactions against additional RBPs, and the formaldehyde crosslinking may allow protein partners of the target RBP and their collective RNA cargos to be retrieved.

During the experiments, two adapters were ligated to the purified RNA fragments. In addition, endogenous (in lysate) or the added RNA ligases (in purified RNA) can join two fragments that are crosslinked together, resulting in chimeras. Subsequent paired end sequencing captures the two ends of, and we developed a pipeline to identify, such chimeras (Fig. 1b). Short-distance pairs indicate single fragments, while long-distance pairs indicate two proximally ligated fragments. Distance distribution between paired-end tags (inclusive) is mostly between 100 and 400 nts (Fig. 1c, left side panels, see complete data in Supplementary Fig. 1). In addition, discrete clusters of long-distance reads are detected up to 10 kb for most proteins,

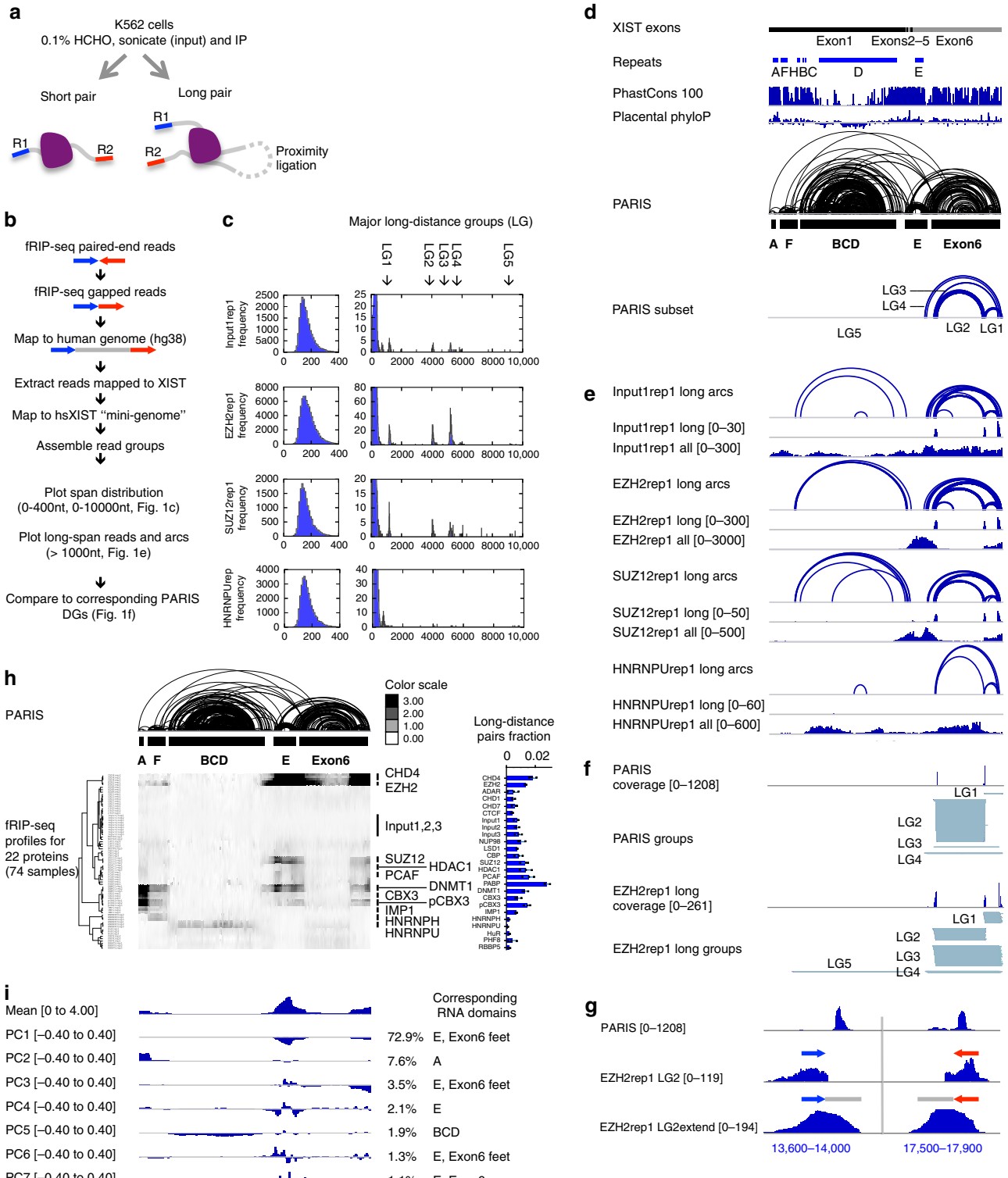

including input control (Fig. 1c, right panels). Five major long-distance groups (LGs) were identified in XIST. The discrete distribution suggests that the ligation reactions are highly specific for certain positions along the XIST RNA dictated by spatial proximity.

To determine the nature of these LGs, we compared them to the PARIS-derived XIST structure[27] (Fig. 1d). The first four major LGs are mapped to the Exon6 domain, primarily among three anchor points, while LG5 is mapped to the large BCD domain (Fig. 1e). LG1–4 overlap RNA duplexes from PARIS

(Fig. 1f); the sequencing tags extended to the approximate length of the RNA fragments clearly overlaps the two arms of the PARIS duplex (Fig. 1g). LG5 does not directly overlap duplexes in PARIS but is consistent with the overall shape of the BCD domain. Together, these long-distance crosslinking data support the XIST domain architecture.

**fRIP-seq reveals spatial partition of XIST-associated proteins**. To understand how XIST-associated proteins are assembled, we

**Fig. 1 fRIP-seq confirms XIST domains and reveals modular RNA–protein interactions. a** Schematic diagram of fRIP-seq experiment. Blue and red lines (R1 and R2) represent paired-end sequence tags. Gray lines represent the non-sequenced regions of RNA fragments, each ~200 nt. **b** fRIP-seq analysis strategy. Paired-end reads are mapped to the genome, which reveals non-sequenced fragment as a gap (gray line between R1 and R2). Mapped reads were remapped to the mature XIST. **c** Distribution of gaps between paired-end tags R1 and R2. Most tag-pairs are from one RNA fragment (left side). A small fraction of pairs are far from each other, therefore most likely from two proximally ligated fragments (right side, same distribution, but highlighting the long-distance pairs). One replicate was shown for each protein and the average read numbers and standard deviations are calculated from all biological replicates ($n = 2$ for EZH2 and $n = 3$ for the rest). **d** Human XIST RNA model, including exons, repeats, phylogenetic conservation (PhastCons 100 and Placental PhyloP from UCSC), and HEK293 PARIS data[27]. Groups that correspond to four fRIP-seq LGs were extracted from PARIS data. **e** Long-distance arcs (tag pairs), coverage of long-distance tag pairs, and coverage of all tags for four examples as in **c**. **f** Comparison of the overlapping duplex groups from PARIS and the LGs from EZH2 fRIP-seq. **g** Comparison of PARIS and EZH2 fRIP-seq LG2 and LG2 extended to the average fRIP-seq fragment size (LG2extend [0–194]). Each side shows a 400-nt window. **h** Unsupervised clustering of XIST–protein interaction profiles in 100-nt windows. A total of 74 samples are clustered, excluding the samples STAG2 (non-specific, as determined by ref. [35]) and WDR5 (low read numbers). The bar graph on the right side shows the fraction of long-distance read pairs pulled down each protein relative to total mapped reads (see numbers in Supplementary Fig. 1a). $n = 3$ biologically independent samples. Data are presented as mean values $+/-$ standard deviation in the bar graph. **i** PCA analysis of profiles in 100-nt windows. The numbers on the right are variation explained by each component. Source data are provided as a Source data file.

normalized all 74 fRIP samples (including input) against the average of input controls in 100-nt windows (see "Methods," the 25th percentile set to 0.1 for each fRIP-seq profile), and clustered enrichment profiles (Fig. 1h, see Supplementary Fig. 1 for all tracks). Several patterns emerged, including the selective enrichment in the A-repeat domain (DNMT1, CBX3, and phosphorylated CBX3 (pCBX3)), F domain (DNMT1, CBX3, pCBX3, and HNRNPH), BCD domain (HNRNPU, which is also in the Exon6 domain), and E domain (CHD4, EZH2, SUZ12, DNMT1, and pCBX3), which is coupled to the end of the transcript. The enrichment on the two ends of the Exon6 domain correlate with the higher percentages of long-distance read pairs (Fig. 1h, bar graph, and Supplementary Fig. 1d), providing further support for the bona fide long-range interactions. Together, these data show clear spatial separation of the XIST-associated proteins.

To automatically derive the domain definitions from the high-dimension data, we applied principal component analysis (PCA; Fig. 1i). The first 4 principal components (PCs) account for 86% of all variation, while the first 7 PCs account for >90% of all variation. The major domains are all detectable. For example, PC1 contains the coupled E-repeat and the feet of the Exon6 domain. PC2 primarily shows the different enrichment patterns in A and F domains. PC5 shows the differential enrichment in the largest BCD domain. Together these data not only confirm the XIST RNA structure but also reveal patterns of protein binding on the XIST RNA.

**Modular assembly of the XIST RNP based on eCLIP.** To further understand the organization of the XIST RNP complex, we used published RNA–protein UV crosslinking data (eCLIP) to map the binding sites of XIST-associated proteins[36]. Yeo and colleagues published a large set of RBP eCLIP data that maps the binding sites to nt resolution. The original analysis revealed enrichment of only four proteins on the transcript level, including HNRNPM, HNRNPK, RBM15, and PTBP1, whereas many other XIST-associated proteins did not pass the stringent enrichment threshold. Out of the 81 proteins previously detected by Xist chromatin isolation by RNA purification in mouse cells[22], 27 of them are included in the 121 eCLIP dataset (Supplementary Data 1 and Supplementary Fig. 2, see Supplementary Fig. 3 for all genome browser tracks).

Similar to the fRIP-seq, the existence of background renders the transcript-wide enrichment less obvious. To detect binding sites on XIST, we performed unsupervised clustering and PCA analysis on eCLIP data for 121 RBPs in K562 cells (Fig. 2). Hierarchical clustering of the eCLIP profiles showed a pattern highly similar to the fRIP-seq data (Figs. 1h and 2b). The PCA analysis also revealed a pattern of RNA domains similar to the

fRIP-seq, with slight differences in the intensity of the domains (Figs. 1i and 2c), although the top PCs explained less variation than the fRIP-seq data, likely due to the larger number of profiles (28.5% for eCLIP PC1 vs. 72.9% in fRIP-seq PC1). For example, PC1 corresponds to the proteins that preferentially bind the feet of the Exon6 domain, the same as in fRIP-seq, while PC2 and PC3 corresponds to enrichment primarily on the A and F domains. The middle regions of the BCD and Exon6 domains are depleted of most protein binding. Together, these data not only support the domain architecture of the XIST RNA but also revealed clustered binding of the proteins (Fig. 2d). We named each of the major structural domains of XIST after the primary sequence repeat that is present in the corresponding domain (domains A, F, BCD, E), except the very 3' Exon6 domain that contains no sequence repeats. Salient features for three of the domains are presented below, and a subset of the associated proteins are discussed.

The A-repeat region is essential for the silencing activity of XIST[40]. We have previously found that SPEN specifically binds the A-repeat[27]. In addition, eCLIP and fRIP nominates multiple additional proteins that bind the A-repeat (Fig. 2d, e, left panel, see Supplementary Fig. 2j for RBP enrichment distribution). SPEN and RBM15 and RBM15B are in the same family, each having similar N-terminal RRM domains and a C-terminal SPOC domain. SRSF1, RBM22, and U2AF1 are all directly involved in splicing. Both SRSF1 and U2AF1 showed clear interaction with Exon2, consistent with their possible roles in splicing during XIST biogenesis. However, we did not observe clear interaction with XIST introns, probably due to the efficient splicing that results in most XIST transcripts being the mature form. The folding up of F domain and BCD domain brings the A-repeat region close to the internal exons 2–5, suggesting a role of the high-level architectures in regulation of splicing. Panning and colleagues identified SRSF1 as an essential A-repeat-associated factor for efficient XIST splicing[41]. Our analysis thus provides a potential explanation for how the distant binding at the A-repeat could affect the splicing of the internal exons. Interestingly, all six proteins are crosslinked to the A-repeat in a periodic fashion, primarily in the single-stranded spacer regions at the junction of double-stranded duplexes formed by hybridization of the repeat subunits (Fig. 2e, middle panel, vertical lines). Averaging the repeats showed that the primary crosslinking sites are in the single-stranded spacer region, with slight differences among the six proteins (Fig. 2e, right panel). The high affinity of multiple proteins to the A-repeat domain suggests that this domain acts as a nucleation center for XIST RNP assembly.

Multiple proteins are enriched in the F domain, including the splicing factors U2AF1 and RBM22, and several HNRNP

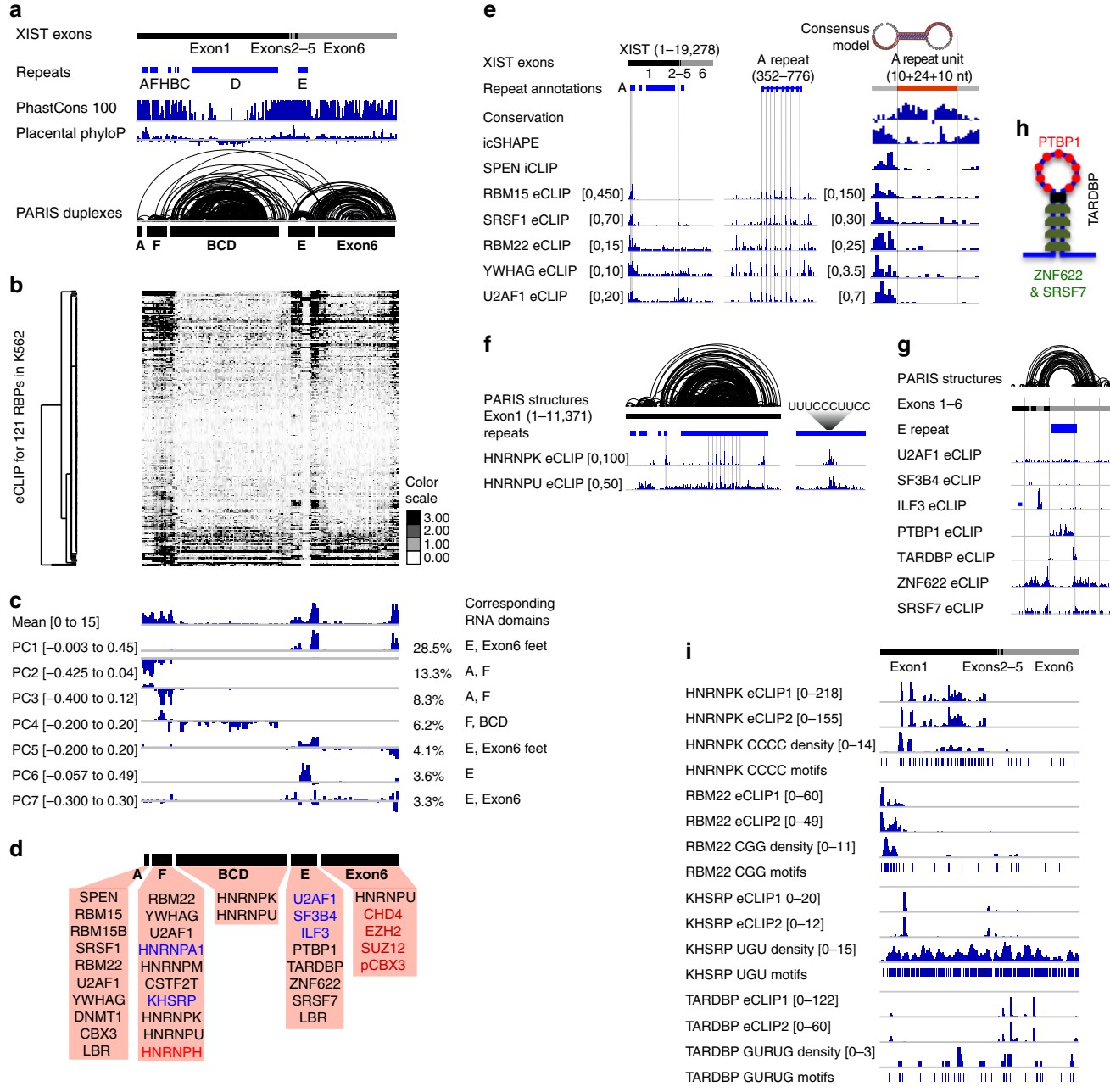

**Fig. 2 eCLIP analysis of XIST-associated proteins reveals modular domains. a** Annotation of human XIST exons, repeats, phylogenetic conservation, and HEK293 PARIS data, same as Fig. 1d. **b** Unsupervised clustering of XIST–protein interactions in 100-nt windows for 121 proteins (242 samples). **c** PCA analysis of eCLIP data. The mean and first seven components are displayed together with variation explained by each. **d** Proteins associated with each domain. Black and blue letters: enriched proteins based on eCLIP. Black: broad and clustered binding. Blue: focal binding. Red: enriched proteins based on fRIP-seq. **e** RBP interactions on the A domain. Left panel: enrichment of 6 RBPs along XIST based on eCLIP. The three vertical lines highlight the A-repeat region and the SRSF1 and U2AF1 peaks on exon 2. Middle panel: zoom-in to the A-repeat domain and the vertical lines indicate protein crosslink positions on the repeat units. Right panel: average signal on a single 24-nt repeat with 10-nt flanking spacers on each side, and the vertical lines mark the start and end of the 24-nt repeat sequence. Conservation, icSHAPE, and SPEN iCLIP data were from ref. [27]. **f** RBP interactions on the BCD domain. Left panel: HEK293 PARIS data, repeats, and eCLIP data are shown for the entire exon 1[27]. Vertical lines mark the BCD domain boundaries and the internal repeats in the D-repeat. Right panel: the average of all D repeats (290 nts per unit) and the consensus HNRNP-binding site. **g** RBP interactions on the E domain. Parts of exons 1 and 6 and the entire exons 2–5 are shown together with HEK293 PARIS data[27]. The vertical lines mark the boundaries of the stem and loop regions. **h** RBP interaction model for the E domain. **i** Comparison of protein-binding sites on XIST based on eCLIP and sequence motifs. eCLIP data are normalized against size-matched input in 100-nt windows. The density profile was calculated based on sequence motifs in 300-nt windows and 50-nt steps. Source data are provided as a Source data file.

proteins. A function for the F domain in X inactivation has not been described. The close proximity to the A and BCD domains may explain the association with similar protein factors.

The middle of the two large domains, BCD and Exon6, are generally depleted of protein binding except HNRNPK and HNRNPU based on the fRIP-seq and eCLIP (Figs. 1h, i and 2b, c, note the lower signal in both mean tracks). HNRNPK binds

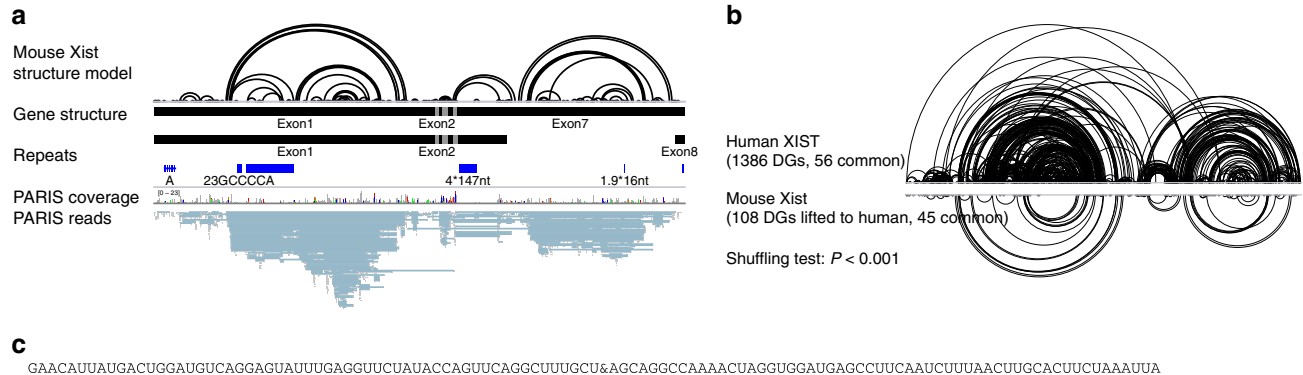

**Fig. 3 Conservation of the XIST architecture in human and mouse. a** PARIS analysis of mouse Xist RNA structure in HATX mES cells. The structure model is the duplexes detected by PARIS. The gene structure tracks are the two mouse mature Xist isoforms. Mouse Xist repeats were detected using the Tandem Repeat Finder[71,72]. **b** Comparison of human and mouse Xist PARIS-determined structures. The mouse Xist DGs are lifted to human coordinates for comparison with the human XIST PARIS data from 293T cells. The *p* value was estimated based on 1000 shuffles, where the overlap of all shuffled structures were below experimentally determined ones (exact *p* value not applicable). **c** A highly conserved long-range duplex structure as detected by PARIS in human and mouse (likely the "boundary element"). Source data are provided as a Source data file.

primarily to the BCD domain to multiple clusters, while HNRNPU binds F, BCD, and middle of Exon6 domains, also in clusters (Fig. 2f, left panel). The human XIST RNA contains 14 units of the D repeat, which explains the periodical binding patterns for HNRNPK and HNRNPU. Averaging eCLIP signal for these two proteins revealed a clear pattern and the pyrimidine-rich consensus sequence (Fig. 2f, right panel). HNRNPU binding is more broader than HNRNPK, especially in the repBCD domain, where the HNRNPU is widely dispersed while HNRNPK is more concentrated on a few peaks. Genetic deletion of mouse Xist BC regions abrogates hnRNPK binding and the associated PRC1 complex, validating this finding[42].

The E domain contains the E repeat, a highly degenerate pyrimidine-rich region, and its surrounding flanking sequences (~600 nts each side). Seven proteins with strong binding sites on the E domain are discovered based on the eCLIP data (Fig. 2g). Two splicing factors, U2AF1 and SF3B4, bind a focal point in Exon2, while the other 5 proteins bind broad regions. ILF3, a double-strand RNA-binding protein bind the evolutionarily conserved Exon4, which is required for high-level XIST expression (Supplementary Fig. 4)[43]. The focal binding of ILF3 to Exon4 is highly significant but was masked by the background when performing whole-transcript enrichment analysis, and therefore was not considered as enriched in the original analysis[36]. PTBP1 binds the E repeat region in the E domain, consistent with its preference for pyrimidine-rich sequences (Fig. 2g, h). TARDBP binds the junctions between the unstructured E repeat and the two stems or the neck of the giant E domain stemloop (Fig. 2g, h). ZNF622 and SRSF7 bind the stem regions, contrary to PTBP1. Together four proteins, PTBP1, TARDBP, ANF622, and SRSF7 show clear spatial partition on the E domain (Fig. 2h).

Similar to the large BCD domain, the center of the Exon6 domain is generally depleted of protein binding. The most significantly associated protein is HNRNPU, which was detected in both fRIP and eCLIP. Interestingly, the two sides, or feet, of the Exon6 domain associated with multiple proteins. This bimodal placement is consistent with the overall structure of the Exon6 domain, which brings the two ends to physical proximity.

**Sequence and structure specificity of RBP binding**. The clustered binding sites that correlate with high-level structural features suggest that RNA structures contribute to binding specificity. To understand how RBP specificity is determined, we analyzed the correlation between sequence motifs and actual binding sites from eCLIP experiments[44,45]. For HNRNPK and RBM22, the eCLIP read density closely follows motif density, while for KHSRP and TARDBP, there is very little correlation, suggesting that additional factors contribute to the RBP specificity (Fig. 2i). We also noticed that multiple proteins are bound to a few regions, in particular the A-repeat domain, the F-domain, the two sides of the E domain, and the ends of the Exon6 domain. These regions are several hundred nts each, thus providing space for multiple proteins to bind simultaneously. XIST can adopt multiple RNA conformations in living cells, and these RBP interactions are ensemble measurements of all XIST molecules in the cell at the same time. It is also likely that some of the proteins cooperate or compete in binding XIST.

**XIST domain architecture conservation in mammals, including mouse**. Using PARIS, icSHAPE, and PARIS-guided multiple sequence alignments, we previously found that the XIST structure domains are highly conserved in evolution, despite limited sequence conservation[27]. To further confirm the structure conservation in mouse, we performed PARIS in mESC line HATX3 ($Xist^{TX/TX} Rosa26^{nlsrtTA/nlsrtTA}$), expressing Xist from the endogenous locus under a tetracycline-inducible promoter[25] (Fig. 3a). Despite the lower sequencing coverage, we detected 108 duplex groups (DGs) after lifting to the human XIST coordinates and found that all the previously discovered major domains are present in mouse Xist. To directly compare XIST structures between human and mouse, we lifted mouse PARIS data to human XIST coordinates (Fig. 3b). About 42% of mouse Xist DGs overlap with human XIST DGs from HEK293 cells[27] (1000 times shuffling of all duplexes, *p* value <0.001), suggesting that the overall Xist structure is conserved between human and mouse, despite the major differences in the size of repeats[46]. Therefore, we conclude that the overall architecture of XIST is conserved in evolution. One of the most highly conserved long-range duplexes, connecting the start and end of the BCD domain, is very stable, with 37 closely stacking base pairs (minimum free energy = −31.70 kcal/mol, Fig. 3c).

**XIST domain architecture determines m⁶A modification specificity**. XIST RNA contains a large number of m⁶A modifications in both human and mouse cells[28,47]. m⁶A sites closely follows RBM15/RBM15B occupancy, which recruit the METTL3/14 methyltransferase complex, suggesting that these two adapter

proteins guide the modification. Patil et al. proposed a model where the location of RBM15/15B proteins determines m6A modification sites. Close examination of CLIP data showed that the RBM15/RBM15B-binding sites are primarily clustered in the A-repeat domain and the other sites are much weaker, raising the question of how m6A is placed at distal locations on XIST RNA over 10,000 bases away. Instead, we noticed that all m6A modification sites, as well as the RBM15/RBM15B-binding sites, are in close spatial proximity when the XIST RNA is folded (Fig. 4a). The consensus m6A motif DRACH is nearly uniformly distributed along the XIST RNA transcript in both human and mouse, with the exception of the pyrimidine-rich E repeat, and at a density of one motif every ~54 nts (Fig. 4a). The discrepancy between m6A motifs and modification sites suggests that the folding of the XIST RNA into compact modular domains contribute to the specificity of m6A modifications. Here we propose two hypotheses to explain the pattern of m6A modifications. First, the compact folding of the large BCD and Exon6 domains exclude the m6A methylase complex. Second, the folding of the XIST RNA creates local proximity among the modification sites with the A-repeat domain. These two possibilities are not mutually exclusive. Altering the overall structure of the XIST RNA is challenging because all the base pairing interactions contribute to the whole transcript structure; disruption of a small number of base pairs is unlikely to cause global changes.

To test these hypotheses, we moved the A-repeat sequence to other locations along the Xist RNA by genome editing and then tested the modification patterns using m6A-RIP-seq. We used the J1 male mESCs with an insertion of doxycycline-inducible promoter (here designated as wild type (WT)), and a derived cell line where the A-repeat region was deleted (ΔSX, removing about 900 nts from the beginning of the transcript)[40] (Fig. 4b). Prolonged induction of WT Xist expression would lead to cell death due to silencing of most genes on the sole X chromosome in these cells, while induction of the ΔSX line did not lead to cell death because absence of the A-repeat region abrogates gene silencing[40]. We moved the 440-nt A-repeat (less than the deleted region, which is larger than the A-repeat alone) to 3 locations, one in the middle of the large BCD domain (knockin at 5136 bp, or KI5), one in the middle of the Exon6 domain (knockin at 14,575 bp, KI14), and one at the end of the transcript (knockin at 17,623 bp, KI17) (Supplementary Fig. 5a). Relocation to the middle of the large BCD and Exon6 domains is likely to induce local modifications near the insertion sites, while relocation to the end of the transcript is likely to induce modifications in physical proximity.

We generated four isogenic A-repeat insertion lines, one for KI5, one for KI14, and two for KI17 using CRISPR/Cas9-mediated gene editing (Supplementary Fig. 5a–c). Then we performed m6A-RIP-seq on all the six cell lines (Fig. 4b). We measured global changes in m6A modifications using m6Aviewer (Supplementary Fig. 5), as well as changes on the XIST RNA alone in custom-defined regions (Fig. 4c, d). In order to compare m6A levels in different Xist alleles, we mapped all RIP-seq reads to the WT sequence, even though in the ΔSX, KI5, KI14, and KI17 Xist alleles, the location of the A-repeat has been altered (Fig. 4d–j).

Four primary m6A domains were defined based on proximity: one surrounding the A-repeat region (m6AD1), two around the E domain (m6AD2 and D3), and one at the end of the transcript (m6AD4). In addition, we also quantified the modification near the insertion sites (m6AKI5 and m6AKI14). WT mouse Xist harbors m6A modification sites in a pattern very similar to the human XIST (compare Fig. 4a and Fig. 4d), and these modification sites are located at the feet of the large RNA domains, whereas the internal regions are almost completely depleted of m6A modifications, despite the presence of m6A motifs. Removal of the A-repeat greatly reduced m6A

modification along the entire Xist RNA except for m6AD3, suggesting that the A-repeat is largely required for modifications at distant regions (Fig. 4d–k). The residual modifications are likely due to the inherent ability of these regions to recruit the m6A methylase independent of the A-repeat region.

Insertion of the A-repeat in the middle of the large BCD domain induces modifications near the insertion site (Fig. 4f) but did not change modification levels at other locations (Fig. 4e, g–j). Similarly, insertion in the middle of the large Exon6 domain induces local modifications without affecting other regions (Fig. 4i, compared to Fig. 4e–h, j). The modifications at these two insertion sites, KI5 and KI14, suggest that the sequences in the middle of the large domains are indeed receptive to modifications, and their lack of modifications in the WT suggest exclusion of the methylase complex (we cannot analyze the exact modified residues due to the lower resolution of RIP-seq). Insertion at the end of the transcript (KI17) led to increases in modifications at all four primary m6A domains, m6AD1–4, without affecting the internal regions of BCD and Exon6 domains (m6AKI5 and m6AKI14). The 5′end of the transcript is also modified at higher levels upon A-repeat insertion at the end of the transcript. These data support both hypotheses that the large domains are excluded from modifications and that regions in physical proximity are modified upon folding of the RNA (Fig. 4k, l). The XIST RNP can be visualized as a splayed-out hand: The A-repeat is the thumb. Moving the A-repeat thumb to the tip of any finger locally affects just that finger. Moving the thumb on the contralateral side of RNA hand restores spatial proximity and m6A modification to the base of the RNP hand (Fig. 4l).

### Spatial separation XIST functions in binding chromatin and nuclear lamina.
During XCI, the HNRNPU family proteins and CIZ1 tether the XIST RNP to the inactive X chromosome, while the LBR protein tethers the XIST RNA to the nuclear lamina[30–34]. Together, the XIST RNP complex acts as a bridge to bring the X chromosome to the nuclear periphery for remodeling and silencing. To understand how the multiple XIST-associated proteins coordinate the localization of Xi and silencing, we used PIRCh sequencing[37] to identify regions of RNA that associate with chromatin (Fig. 5a) and analyzed the binding sites of these proteins on XIST (Fig. 5c).

fRIP-seq and eCLIP experiments showed that the HNRNPU, CIZ1 (based on PCR), and LBR-binding sites are distributed along the XIST RNA (Figs. 1h and 2, summarized in Fig. 5c). In particular, HNRNPU bind the bodies of the large domains, while LBR is enriched on the A-repeat domain and the feet of the larger domains (Fig. 5c). The HNRNPU enrichment profiles from fRIP-seq and eCLIP are highly consistent (Fig. 5d), while the HNRNPU fRIP-seq and LBR CLIP profiles show significant anti-correlation (Fig. 5e). We hypothesize that XIST regions that are bound by HNRNPU would be more tightly associated with the chromatin, and such regions can be crosslinked to chromatin using glutaraldehyde, a non-specific and highly efficient cross-linker of macromolecules that contain nucleophilic groups like primary amines. After crosslinking, we used sonication to fragment chromatin to small pieces and enriched chromatin-associated RNA using antibodies for histones. The purified RNA were then sequenced to determine relative enrichment (Fig. 5a). Interestingly, we found that the chromatin-associated regions are primarily in the large domains associated with HNRNPU and CIZ1 (Fig. 5c), confirming the spatial separation of XIST domains in binding chromatin and nuclear lamina. The enrichment of the C-repeat region by PIRCh is consistent with previous report that showed a role of the C-repeat in chromatin binding[48].

In previous studies, it was noted that LBR binds three discrete regions in mouse Xist (around A-repeat domain and flanking the

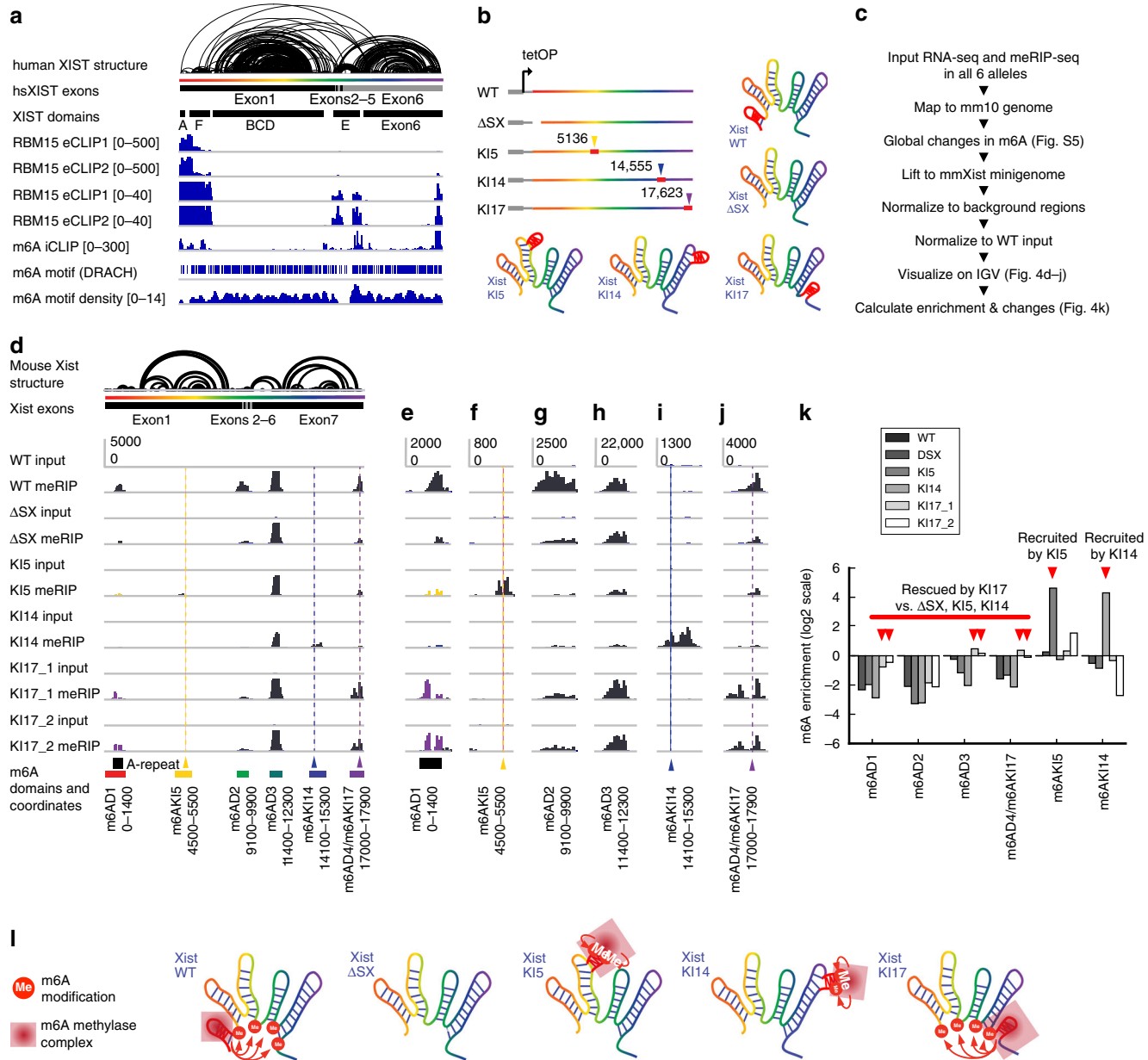

**Fig. 4 XIST RNA structure determines m6A modification patterns. a** m6A on the human mature XIST RNA. K562 RBM15 eCLIP was normalized against input in 100-nt windows[36] and plotted in two scales to highlight the differences in binding the 5′ end and other regions. HEK293 cell m6A iCLIP track was from ref. [47]. m6A motif density was calculated in 300-nt windows and 50-nt steps. **b** Gene structure for the alleles. WT and ΔSX (~900 bp deletion in Xist 5′ end) alleles were under the control of tetracycline-inducible promoter. A-repeat relocation alleles KI5, KI14, and KI17 were derived from ΔSX by A-repeat insertion in the indicated locations. **c** MeRIP-seq analysis pipeline. Global analysis of m6A changes was performed on data mapped to the mm10 genome, while targeted analysis was performed on the mature mouse Xist transcript. **d** m6A sites are changed after relocating the A-repeat domain. The mouse Xist RNA PARIS model is the same as in Fig. 3a. m6A domains are labeled under the genome browser tracks. *Y*-axis is the same in each track. All data including the A-repeat relocation alleles were mapped to the same wild-type Xist mature RNA. **e–j** Zoom-in view of all m6A domains. Location of the original A-repeat is indicated in **e**. The A-repeat knockin locations are indicated in **f**, **i**, **j**. *Y*-axis scales are the same for all tracks. **k** Quantification of m6A changes relative to wild type in log scale in pre-defined m6A domains shown in **d**. One replicate was available for each sequencing library. **l** A mechanistic model of RNA structures in guiding m6A modifications. The A-repeat domain recruits the m6A methylase complex to modify sequences physically close to the domain. The residual modification on Xist after A-repeat deletion was due to its intrinsic ability to recruit m6A methylase complex. Relocation of the A-repeat to the inside of the large domains (KI5 and KI14) induces local modifications (m6AKI5 and m6AKI14). Relocation of the A-repeat to the end of the transcript (KI17) induces modification in physical proximity (m6AD1, m6AD2, m6AD3, and m6AD4). Source data are provided as a Source data file.

E domain)[29]. Deletion of the A-repeat region reduced LBR binding to the latter two sites, suggesting cooperative binding. These data suggest that sequence alone is insufficient to determine LBR binding. In particular, we found strong anti-correlation for LBR binding and PIRCh enrichment in the E-domain (Fig. 5f). This pattern is identical to the spatial separation

of RBP binding based on eCLIP studies (Fig. 2g, h). In light of the XIST structure model, it became clear that the cooperative binding of LBR to three distant locations would be mediated by the physical proximity of the folded XIST RNA. Here the A-repeat domain likely serves to bring LBR to the other locations in physical proximity. We hypothesize that the physical proximity of

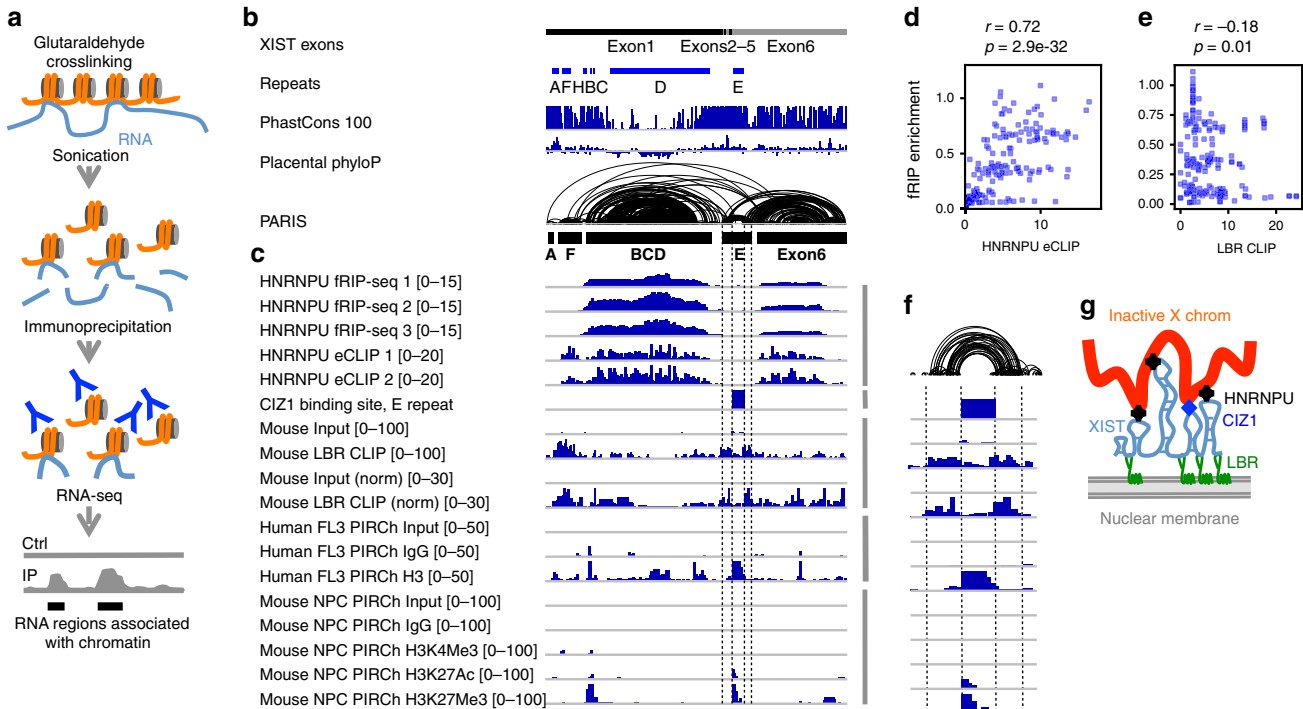

**Fig. 5 Spatial separation of XIST RNP functions in binding chromatin and nuclear lamina. a** Schematic diagram of the PIRCh method. Orange lines: genomic DNA, light blue lines: RNA. The Y shape represents antibodies against histones. Ctrl control, IP immunoprecipitation. **b** Annotation of the human XIST RNA. XIST exons and repeats, phylogenetic conservation (PhastCons 100 and Placental PhyloP from UCSC), and PARIS data in HEK293 cells are shown, same as Fig. 1d. **c** HNRNPU fRIP-seq and eCLIP data in human HEK293 cells were normalized against their input controls in 100-nt binds. The mouse CIZ1-binding site track was made based on Ridings-Figueroa et al.[32] and Sunwoo et al.[33], showing the enrichment of CIZ1 binding on the E-repeat. The LBR raw data and normalized enrichment ratios (in 100-nt bins) were from mouse ES cells (Chen et al.[29]) and then lifted to human XIST coordinates. Human and mouse PIRCh data were all normalized against their own controls, respectively, in 100-nt bins. The mouse PIRCh data were lifted to human XIST coordinates. **d** Correlation of HNRNPU eCLIP and fRIPseq enrichment in 100-nt bins. Spearman's rank correlation was calculated and the raw *p* values were provided. **e** Anti-correlation of HNRNPU fRIP-seq and LBR CLIP enrichment in 100-nt bins. **f** Zoom in view of the E-repeat domain, showing the anti-correlation of CIZ1 binding vs. LBR CLIP and LBR binding vs. PIRCh. **g** Model for the spatial separation of XIST RNP functions in binding the inactive X chromosome and the nuclear lamina, color coded the same way as **a**. Source data are provided as a Source data file.

the A-repeat domain to the feet of the other domains is required for cooperative LBR binding to XIST (Fig. 5g).

To test the role of A-repeat as a nucleation center, we performed infrared CLIP (irCLIP) on mESCs expressing the A-repeat relocation alleles (Fig. 6a). A-repeat deletion (ΔSX) reduced LBR binding across Xist. Insertions of the 440-nt A-repeat at 5 and 14 kb not only resulted in the binding at transplanted A-repeats but also at sequences near insertion sites, suggesting that the A-repeat is able to recruit protein binding to proximal regions (Fig. 6a–h). More interestingly, A-repeat insertion at the transcript 3′ end (KI17) induced binding to both the 3′ end and m6AD3 that are not close in sequence but are in spatial proximity. These data demonstrated that sequence alone was insufficient for protein binding, A-repeat serves as a nucleation center, and the folding of the XIST RNA serves as a conduit for recruiting protein binding to physically close regions (Fig. 6i). We also performed irCLIP on SPEN, which serves as a bridge to bring the HDAC complex to Xist[2]. Again, we found that the insertions in the middle of the large domains (KI5 and KI14) resulted in local spreading of SPEN near the A-repeat insertion sites, similar to what we observed for m6A modification and LBR binding. The KI17 allele showed spreading around the 17-kb insertion site and modest enhancement of binding at m6AD3. Most distant sites such as m6AD1 and m6AD2 were not affected.

The variable levels of rescue for m6A (Fig. 4) or RBP enrichment (Fig. 6) levels at distant sites could be due to several reasons. First, there is likely difference in the affinity of antibodies

used for the immunoprecipitation of m6A, LBR, and SPEN. Second, the mechanisms of recruitment are different for each of the three interactions. For example, m6A is deposited by the WTAP-Mettle3-Mettl14 complex which is recruited by RBM15 and other associated proteins that associate with the A-repeat; SPEN is large protein that directly binds the A-repeat, while the tight association of LBR to the nuclear membrane LBR limits its mobility. Despite these differences, the consistent ability of the relocated A-repeat to recruit m6A modification and protein binding to the sequences around the insertion sites and in physical proximity provides strong evidence for the role of XIST RNA structure in organizing the overall RNP structure functions.

Quantitative reverse transcription PCR analysis showed that each of the KI alleles accumulates lower Xist RNA level than WT and is unable to induce silencing of X-linked genes (Supplementary Fig. 6). The A-repeat relocation results in multiple types of effects in addition to changing the m6A modification sites. For example, LBR binding is altered, potentially changing the XIST interaction with the nuclear membrane; SPEN binding is also altered, potentially changing the recruitment of the HDAC3 complex. The other proteins that bind the A-repeat are probably also altered to some extent, including the splicing factors such as SRSF1, U2AF1, and RBM22. These aspects are all inter-related in XIST repressing X chromosome genes. For example, both m6A and splicing could affect XIST stability. XIST levels, m6A modification, HDAC recruitment, and LBR-mediated nuclear membrane binding all affect the silencing activity. We also

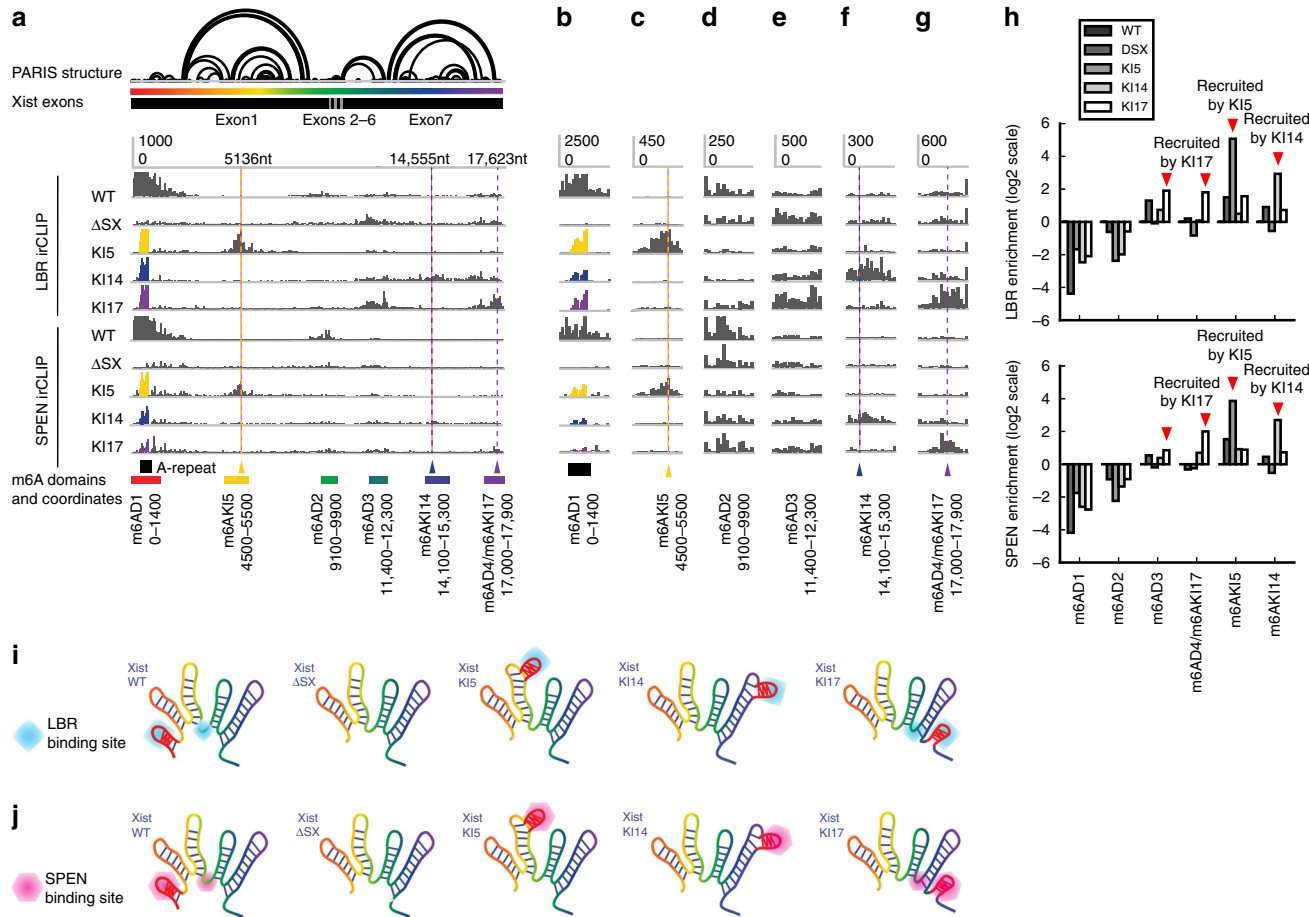

**Fig. 6 XIST architecture regulates protein-binding specificity. a** irCLIP of LBR and SPEN in mES cells expressing XIST alleles with relocated A-repeat. The mouse Xist RNA PARIS structure model is the same as in Fig. 3a. The m6A domains are labeled under the genome browser tracks. The Y-axis is the same in each track. All data including the ones with A-repeat insertion at other locations were mapped to the same mature wild-type Xist RNA. **b**–**g** Zoom-in view of all the domains as defined for m6A. Location of the original A-repeat is indicated in **b**. The A-repeat knockin locations are indicated in **c**, **f**, **g**. Y-axis scales are the same for all tracks in each panel. **h** Quantification of the changes of protein binding relative to wild type in log scale in the pre-defined m6A domains shown in **a**. One replicate was available for each sequencing library. **i**, **j** Models of the role of RNA structures in guiding protein binding. The A-repeat domain recruits its associated proteins to sequences that are physically close to the domain. The residual binding on Xist after A-repeat deletion was due to its intrinsic ability to recruit proteins (see the ΔSX tracks in **d**–**j**). Relocation of the A-repeat to the inside of the large domains (KI5 and KI14) induces local binding (m6AKI5 and m6AKI14). Relocation of the A-repeat to the end of the transcript (KI17) binding in physical proximity (m6AD3 and m6AD4). Source data are provided as a Source data file.

acknowledge that the current A-repeat relocation alleles, with the variable expression level, are not perfect for the purpose of dissecting the relative contribution of these different pathways in XCI. The lack of obvious functional consequences is likely the result of lower XIST expression and other factors. A recent study showed that the A-repeat itself contributes to the promoter activity[49], which explains why the deletion of A-repeat resulted in lower expression. More quantitative analysis of the A-repeat relocation alleles that express XIST at identical levels is needed in future studies.

## Discussion

Long RNAs, including the protein-coding mRNAs and lncRNAs, make up the majority of the transcriptome. The dynamic interactions and structures of these RNAs and their protein partners are essential for the exquisite control of gene expression, yet pose major challenges for structure and function analysis. Most of these large RNP complexes are heterogeneous and contain many weak interactions and therefore cannot survive the harsh conditions of purifications for in vitro analysis by crystallography, nuclear magnetic resonance (NMR), and cryo-electron microscopy.

Building upon the PARIS method[27], we integrated multiple approaches for the comprehensive characterization of large RNP complexes, including nt flexibility measurements (e.g., icSHAPE)[50], phylogenetic conservation of structures, crosslink and proximity ligation of protein-bound RNA structures (e.g., chimeric reads in fRIP-seq), and unsupervised clustering and PCA analysis of RBP-binding profiles on RNAs. These orthogonal approaches reveal the organization principles for large RNPs, and their associated functions, from the base pair level to the domain level (hundreds to thousands of nts). Proximity ligation is the basis of PARIS, and chimeric reads between microRNAs and mRNAs have previously been used to pinpoint microRNA targets[51,52]. In this work, the crosslink and proximity ligation principle that has been successfully employed in the analysis of chromatin structures and RNA interactions and structures can be extended to the analysis of other RBPs on any RNA of interest, as long as these proteins are crosslinkable. The application of these methods will be instrumental in the analysis of other RNP complexes. More importantly, the discovery of compact RNP domains set the stage for focused in vitro studies of these domains through purification reconstitution and structure analysis using physical methods[53].

Multiple previous studies have analyzed the XIST RNA, either in part or in its entirety, using various methods[18,19,54–57]. Duszczyk et al. determined the in vitro solution structure of a partial A-repeat unit (14 nts out of the 24-nt unit) using NMR and found a stable stemloop structure[56]. Two studies used chemical probing to measure nt flexibility of the in vitro transcribed A-repeat region and found a number of inter-repeat and intra-repeat duplexes[54,57]. These conflicting in vitro models have not been reconciled. Two additional studies used chemical probing in living cells to analyze XIST RNA structures[18,19]; however, these models may have limitations because (1) chemical probing reports whether each nt is base paired or constrained by protein binding and does not directly capture the base pairing relationship and (2) the secondary structure modeling is based on the faulty assumption that only one stable conformation exists and thus misses alternative conformations and long-range structures[20,27].

Using a combination of five orthogonal methods, we have built a comprehensive model of the XIST RNP complex. In our previous study, we have applied PARIS, icSHAPE, and phylogenetic analysis to determine the overall structure of the XIST RNA. In the current study, we incorporated systematic analysis of RNA–protein interactions data based on fRIP-seq and eCLIP and also examined the proximally ligated reads from RNA–protein interactions. Recently, Moore and colleagues analyzed RNA regions associated with the exon junction complex long-distance RNA structures in XIST similar to the PARIS-derived modular domains[55]. These studies firmly established the modular architecture of the entire RNP complex. Sequence inside the module are more likely to base pair with each other and also interact with similar RBPs, while sequences outside of the module may be excluded for interactions.

The repetitive nature of the A-repeat has made it a challenging target for structure analysis. As discussed above, several studies have reported structure models for the A-repeat region using chemical probing methods, leading to conflicting models[2,57]. We have used several methods to establish a stochastic inter-repeat duplex model[27]. Using CLIP and gel shift assays, we found that the A-repeat region forms a multivalent platform to bind the SPEN adapter protein. Importantly, our model has been confirmed by a more recent rigorous phylogenetic analysis of non-coding RNA structure conservation[58]. In the current study, we further extended the model of the essential A-repeat domain by identifying additional potential interactions, including RBM15/RBM15, SRSF1, U2AF1, and LBR. This multitude of high-affinity A-repeat-associated proteins suggests that the A-repeat serves as a nucleation center for recruiting XIST-binding proteins. Using the A-repeat relocation XIST alleles and CLIP experiments, we showed that the A-repeat is indeed sufficient in spreading physically local RBP binding and m6A modifications (Figs. 4 and 6). The nucleation function of A-repeat together with the topology of the entire XIST RNA are responsible for generating the unique patterns of protein binding, as well as the functions associated with these proteins. The multiple protein binding to the A-repeat create a crowded environment, and it remains to be determined how these proteins are assembled in spatial- and temporal-specific manner. For example, the splicing factors are required early on for proper XIST maturation, while the effectors of XIST functions may bind XIST later. It has been shown that XIST interactions with proteins change during stem cell differentiation[22]. It is conceivable that a dynamic process of XIST RNP assembly coordinates its functions. In addition, other ways of organizing the functions are also possible.

XCI is a complex process and XIST coordinate multiple steps in this process. Previous studies have discovered distinct regions in XIST that performs separate roles, suggesting a modular organization of functions. For example, the A-repeat was first found to be required for recruiting the SMRT-HDAC3 complex to repress transcription[24–27]. Several discrete regions are involved tethering XIST to the nuclear membrane by the LBR protein[29]. The attachment to the inactive X chromosome is mediated by HNRNPU family proteins and CIZ1 that bind other discrete regions[30–34]. However, the previous coarse-grained deletion studies cannot clearly define the domains. The discrete regions in XIST that coordinate the same functions are particularly difficult to visualize. Here we show that there is no strict correlation between the higher-order RNP domains with the repeats. For example, the F domain is much larger than the F repeat alone; all three BCD repeats are folded into the same domain, together with non-repetitive sequences; the E repeat domain includes both the E-repeat and surrounding sequences; the large Exon6 domain does not contain extensive repetitive elements. Therefore, one cannot deduce the structural modularity from these deletion studies alone.

In the present study, we have found that the XIST-associated functions are spatially separated on the RNA structural scaffold. For instance, the A-repeat domain together with the physically close regions of the RNA located at the feet of the other domains associate with proteins involved in transcriptional silencing, m6A modification, splicing regulation, DNA methylation, and nuclear lamina attachment. The body of the large BCD, E, and Exon6 domains binds HNRNPU family proteins and CIZ1, which then tethers XIST to the inactive X chromosome. The mechanisms that coordinate other functions in XIST remain to be discovered. The discovery of the modular domain architecture provides a framework for future analysis of other functions coordinated by XIST (see Supplementary Discussion).

Previous studies on the yeast 1.2-kb long telomerase RNA revealed an RNA scaffold that contains several essential protein-binding sites[12]. The arms of the RNA scaffold can be relocated without affecting their functions. Thus the individual domains in TERC RNA are like words on a billboard; their presence rather than exact order convey most of the message. Here our studies revealed a more complex picture for the flexible domain architecture of XIST: the scaffold serves to insulate certain regions and bridge other regions. The role of structure-induced proximity in guiding RBP binding and m6A modification is similar to the concept of chromatin conformations guiding gene expression regulations. Chromatin structures can either insulate regions from surrounding epigenetic environment or induce spatial proximity to bring regulatory elements to gene promoters. For example, the structure of the X chromosome guides the spreading of XIST to spatially close sites[59]. The specific order of domains in XIST suggests the existence of grammar rules in lncRNA function. The nature of this grammar, whether for stable RNP assembly or RNP function, remains to be determined.

## Methods

**mESC culture**. Male inducible TXY WT, TXY:ΔSX lines (gift from Anton Wutz)[40], and all generated TXY knockin cell line derivatives (TXY:KI5, TXY:KI14, TXY:KI17) were cultured and treated for 48 h with 4 µg/ml doxycycline before RNA collection. HATX3 cells (Xist[TX/TX] Rosa26[nlsrtTA/nlsrtTA]) were grown under the same conditions before AMT crosslinking and RNA collection[25]. All mESCs were maintained on 0.2% gelatin-coated plates at 37 °C with mES media, which was changed daily: Knockout Dulbecco's modified Eagle's medium + 10% fetal bovine serum + 1% MEM NEAA + 1% GlutaMax + 1% Pen–Strep + 0.2% BME and 0.01% LIF.

**Analysis of fRIP-seq data**. To determine the interactions between XIST and associated proteins, fRIP-seq experiments on 24 chromatin-associated and traditional RNA-binding proteins in human K562 cells were reanalyzed (GSE67963)[35]. In the fRIP-seq experiments, formaldehyde crosslinked RNP complexes were sonicated so that the associated RNA fragments are around a few hundred nts, and the sequenced fragments are around 150 nts (see Figs. 1 and S1 for size

distribution). Paired end sequencing was performed on the libraries, 31 nts each end. The general pipeline is as follows: Convert paired end reads to gap reads → map to hg38 → extract reads mapped to XIST → map to hsXIST "minigenome" → assemble mapped Aligned and Chimeric reads → extract long pairs → make distance distribution, bedgraph, and arcs for both short and long pairs. See supplementary Note 1 for details.

**Analysis of eCLIP data**. The Yeo laboratory performed large-scale CLIP experiments to determine the transcriptome-wide binding sites of >100 proteins in female K562 cells and male HepG2 cells[36] and reported four proteins that bind XIST specifically (>2-fold enrichment). To determine the interactions between XIST and associated proteins, the eCLIP data were reanalyzed as follows. The total numbers of bigwig files used are 726 and 618, respectively, and the files are named as follows K562/HepG2_RBP_0/1/2_neg/pos.bw. See Supplementary Note 2 for details.

**Analysis of RBP motifs based on eCLIP**. RBP motifs were derived from previous publications. These motifs are readily available from previous publications and mapped to the human XIST RNA: HNRNPK (CCCC), KHSRP (CCCC), TARDBP (GTRTG), and RBM22 (CGG)[44,45].

**Analysis of PIRCh data**. See Supplementary Notes 3 and 4 for details on the analysis of mouse NPC and human FL3 PIRCh data.

**PARIS experiments in mESCs**. The PARIS protocol was performed as previously described with slight modifications[27]. HATX mESCs were treated with 0.5 mg/ml AMT (Sigma) and crosslinked with 365 nm UV for 30 min in Stratalinker 2400. Cell lysate was digested with S1 nuclease and RNA purified using TRIzol and further fragmented with ShortCut RNase III. RNA was separated by 10% native polyacrylamide gel and then the first dimension gel slices were further electrophoresced in a second-dimension 20% urea-denatured gel. Crosslinked RNA above the main diagonal was eluted and proximity ligated with T4 RNA Ligase I. After ligation, samples were denatured and purified using Zymo RNA Clean & Concentrator and photo-reversed with 254 nm UV for 5 min. Proximity-ligated RNA molecules were then ligated to barcoded adapters and converted to sequencing libraries. Libraries were sequenced on the Illumina NextSeq.

**Analysis of mouse PARIS data**. See Supplementary Note 5.

**Generation of knockin cell lines**. To test the role of the mouse *Xist* architecture in m6A modification specificity, the A-repeat domain was relocated to several regions in the mouse *Xist* gene using the CRISPR-Cas9 system. Synthesized, high-performance liquid chromatography-purified single-guide RNAs (sgRNAs) were purchased through Synthego. Donor plasmids were generated through overlap extension PCR of 3 fragments: the A-repeat sequence flanked by two 800 bp homology arms to each respective genomic region. The resulting PCR product was then ligated into the pCR-Blunt-II-TOPO vector, using the Zero Blunt TOPO PCR Cloning Kit according to the manufacturer's instructions. Cas9 protein with NLS (PNA Bio) was complexed with sgRNAs in microfuge tubes for 10 min at 37 °C and immediately transferred to ice. In all, $1 \times 10^6$ TXY:dSX cells were nucleofected with pre-complexed CRISPR RNP and 20 µg of donor plasmid using an Amaxa nucleofector with the manufacturer's recommended settings for mESCs. Cells were then grown for 72 h before low-density splitting into single-cell colonies. Colonies were picked using a light dissection scope and grown in 48-well plates to establish clonal populations. For genotype screening, genomic DNA was extracted using QuickExtract (Lucigen) and then subject to PCR with primers spanning the knockin site (see Supplemental Table for details). Clones were screened by looking for a PCR product significantly higher in size than in that of non-targeted TXY: ΔSX cells. PCR products were then Sanger sequenced through Stanford ELIM using the forward PCR primer as a sequencing primer to verify knockin of the A-repeat.

To make the guide RNAs, DNA template was designed as follows: T7_promoter + sgRNA + scaffold (GAAATTAATACGACTCACTATAGG [sgRNA] GTTTTAGAGCTAGAAATAGCAAGTTAAAATAAGGCTAGTCCGTTATCAAC TTGAAAAAGTGGCACCGAGTCGGTGCTTTT). The following single-strand DNA were ordered from IDT DNA and used to make the duplex DNA for in vitro transcription.

mmX.5136f (middle of BCD domain):
GAAATTAATACGACTCACTATAGGGAGTTAGAAAGATGTGACCTGGTTTT AGAGCTAGAAATAGCAAGTTAAAATAAGGCTAGTCCGTTATCAACTTGA AAAAGTGGCACCGAGTCGGTGCTTTT

mmX.5136r (middle of BCD domain):
AAAAGCACCGACTCGGTGCCACTTTTTCAAGTTGATAACGGACTAGC CTTATTTTAACTTGCTATTTCTAGCTCTAAAACCAGGTCACATCTTTCTA ACTCCTATAGTGAGTCGTATTAATTTC

mmX.14575f (middle of Exon6 domain):
GAAATTAATACGACTCACTATAGGTAAAGCCGGGACCTAACTGTGTT TTAGAGCTAGAAATAGCAAGTTAAAATAAGGCTAGTCCGTTATCAACTT GAAAAAGTGGCACCGAGTCGGTGCTTTT

mmX.14575r (middle of Exon6 domain):
AAAAGCACCGACTCGGTGCCACTTTTTCAAGTTGATAACGGACTAGC CTTATTTTAACTTGCTATTTCTAGCTCTAAAACACAGTTAGGTCCCGGCT TTACCTATAGTGAGTCGTATTAATTTC

mmX.17623f (after Exon6 domain):
GAAATTAATACGACTCACTATAGGTATGTGATCAAAGCAGATGAGTT TTAGAGCTAGAAATAGCAAGTTAAAATAAGGCTAGTCCGTTATCAACTT GAAAAAGTGGCACCGAGTCGGTGCTTTT

mmX.17623r (after Exon6 domain):
AAAAGCACCGACTCGGTGCCACTTTTTCAAGTTGATAACGGACTAGC CTTATTTTAACTTGCTATTTCTAGCTCTAAAACTCATCTGCTTTGATCACA TACCTATAGTGAGTCGTATTAATTTC

The following PCR primers were used for cloning and testing (LHA: left homology arm, RHA: right homology arm):

5KI_LHA_F: GAGAAAGCTTGACTTCCAGAGACATAGAATTTCACTTTG

5KI_LHA_R: CCCCGATGGGCAAGAATATATAAACAATGAAGGGCGAT AGCACCCATGAC

5KI_repA_F: TGTCATGGGTGCTATCGCCCTTCATTGTTTATATATTCTT GCC

5KI_repA_R: ATCTCCATCAGTTAGAAAGATGTGACCTGACTCACAAAA CCATATTTCC

5KI_RHA_F: GGTGGATGGAAATATGGTTTTGTGAGTCAGGTCACATC TTTCTAACTG

5KI_RHA_R: GAGAGAATTCTACAAATAAGTCTTCACCAGATG

14KI_LHA_F: GAGAAAGCTTTGCCCAGGTCACATTATG

14KI_LHA_R: CCCCGATGGGCAAGAATATATAAACAATGAAACAGTT AGGTCCCGGCTTTATAG

14KI_repA_F: GTTCTATAAAGCCGGGACCTAACTGTTTCATTGTTTATA TATTCTTGCC

14KI_repA_R: AGAAAGTAATCACTGTTCACTGATAAAGCCAACTCACA AAACCATATTTCC

14KI_RHA_F: GGTGGATGGAAATATGGTTTTGTGAGTTGGCTTTATCA GTGAACAG

14KI_RHA_R: GAGAGAATTCTATATAATTCTTTAAAAATATTATTCAC TCAG

17KI_LHA_F: GAGAAAGCTTTCCTTACTATAATATACTCAAGGTGG

17KI_LHA_R: CCCCGATGGGCAAGAATATATAAACAATGAATGGTAGG ATGTGCTTAATTG

17KI_repA_F: ATATTGCTACCAATTAAGCACATCCTACCATTCATTGTT TATATATTCTTGCC

17KI_repA_R: GTACACAGTTCATTTATGTGATCAAAGCAGATGAACTC ACAAAACCATATTTCC

17KI_RHA_F: GGTGGATGGAAATATGGTTTTGTGAGTTCATCTGCTTT GATCACATAA

17KI_RHA_R: GAGAGAATTCAGGGCCACTGAGTTAGAAAC

Genotyping primers

5KI_Genotype_F and 5KI_Genotype_R: CCAGCCCTGTGTGCATTTAG, AGCCTTATCCAGTGTCCAGG

14KI_Genotype_F and 14KI_Genotype_R: TTCCACCTCCTCAGTCAAGC, TGCTTTGGTGAGGCTCAGTA

17KI_Genotype_F and 17KI_Genotype_R: AGCAAGCCTGACCCTAAAGT, TGGTGGGAAGATGACTCCAG

sgRNA test primers

5KI_gRNA_F and 5KI_gRNA_R: CCAGCCCTGTGTGCATTTAG, GGTTTG ATTCCCCAGCACAG

14KI_gRNA_F and 14KI_gRNA_R: GCCTGGTGTGCAATGACTTT, TGGGA TTATCTCACTCTGGCC

17KI_gRNA_F, and 17KI_gRNA_R: GAGCCAGGTTGAAGAGGTCT, AAC TACCCCACCCACTCAAC

**meRIP-seq in mESCs**. Total mES RNA was subjected to one round of Poly(A)Purist MAG treatment to enrich for polyadenylated RNAs as per the manufacturer's instructions (Ambion). RNA was then fragmented to 100-nt median-sized fragments using RNA Fragmentation Reagents (Ambion) and subjected to one round of m6A immunoprecipitation. For immunoprecipitation of RNA, 5 µg of m6A antibody (Millipore) was coupled to 40 µl Protein A Dynabeads (Novex) in 100 µl 1× IPP Buffer (50 mM Tris-HCl, pH 7.5; 150 mM NaCl; 0.1% NP-40; 5 mM EDTA) overnight at 4 C. Beads were then washed twice in 1× IPP Buffer. Fragmented RNA was denatured at 70 °C for 2 min, cooled on ice, and bound to antibody beads in 185 µl 1× IPP Buffer for 3 h at 4 °C. Beads were then washed sequentially with (2×) 500 µl 1× IPP Buffer, (2×) 500 µl Low Salt Buffer (0.25× SSPE; 1 mM EDTA; 0.05% Tween-20; 37.5 mM NaCl), (2×) 500 µl High Salt Buffer (0.25× SSPE; 1 mM EDTA; 0.05% Tween-20; 137.5 mM NaCl), (1×) 500 µl TET Buffer (10 mM Tris-HCl, pH 8.0; 1 mM EDTA; 0.05% Tween-20). Beads were eluted with 50 µl RLT Buffer (Qiagen RNeasy Mini Kit) and incubated at 25 °C for 5 min and recovered with the RNeasy Mini Kit followed by concentrating with Zymo RNA Clean & Concentrator in 10 µl water. Ten nanograms of input RNA (before immunoprecipitation) and 10 ng of immunoprecipitated RNA were then used to prepare sequencing libraries using the SMARTer Stranded RNA-Seq Kit—Pico Input Mammalian (Clontech #634411), as per the manufacturer's instructions. Sequencing libraries were pooled and sequenced

# ARTICLE

on the Illumina MiSeq (files named *mar08* and *apr26*) and NextSeq (files named *jun24*).

See Supplementary Note 6 for details of the global analyses of meRIP-seq data and targeted analysis of Xist m6A modification.

**Analysis of miCLIP data from Linder et al.**[47]. The miCLIP bedgraph files in GSE63753 were downloaded from GEO and lifted to hg38 using the liftOver tool from UCSC genome brower[47]. Then the data were lifted to the mature XIST transcript coordinates (without introns) using custom python scripts.

**irCLIP analysis of SPEN and LBR.** The irCLIP experiments were performed as described previously[60]. Briefly, mESCs with engineered Xist mutations were cultured with standard conditions and induced to express Xist (see earlier description on cell culture). Afterwards, cells were lysed, immunoprecipitated with antibodies for these RBPs and treated with S1 nuclease. RNP complexes were resolved on denatured polyacrylamide gels and regions above the protein size were excised for RNA extraction and library preparation.

Sequencing output reads were processed by bbmap to remove duplication on fastq level. Remained reads were trimmed off the 3' solexa adapter and against sequencing quality q20 by cutadapt (version 2.4). Trimmed reads were mapped first to RNA biotypes with high repetitivity by bowtie2 (version 2.2.9) to our custom-built indexes: rRNAs (rRNAs downloaded from Ensembl GRCm38.p6/mm10 and a full, non-repeat masked mouse rDNA repeat from GenBank accession No. BK000964), snRNAs (from Ensembl GRCm38.p6/mm10), miscRNAs (from Ensembl GRCm38.p6/mm10), tRNAs (from UCSC table browser GRCm38.p6/mm10), RetroGenes V6 (from UCSC table browser GRCm38.p6/mm10), and RepeatMasker (from UCSC table browser GRCm38.p6/mm10). Remained reads were mapped to mouse genome GRCm38/mm10 by STAR (version 2.7.1a) with junction file generated from mRNAs and lncRNAs by Genocode GRCm38.p6/mm10 GTF file. Only reads uniquely mapped to the mouse genome were included in the downstream analysis. The RBP-binding loci as suggested by the irCLIP method, was defined as 1-nt shift to the 5' end of each mapped read. Each locus was extended 5 nts upstream and downstream to shape a local interval; only intervals overlapped between two replicates were included. Then five nts were trimmed from each side of the overlapped interval to shape the final cluster. Cluster annotation was processed against the Genocode GRCm38.p6/mm10 GTF file. Reads annotated to Xist gene were re-mapped to the Xist mini-genome. Normalization on Xist was processed in the same way as for m6A data.

**Quantification and statistical analysis.** In the relevant figures, figure legends denote the statistical details of experiments, including statistical tests used, kind of replicates, and the value of *n*. Asterisks define degree of significance as described in the figure legends. All Student's *t* test and Mann–Whitney *U* test were analyzed as two sided. All the sequencing data were aligned to mouse and human genomes (mm10 and hg38) or custom-made mini-genomes (mmXist and hsXIST). Statistical analyses and graphics were performed using Python, R, and Microsoft Excel.

**Reporting summary.** Further information on research design is available in the Nature Research Reporting Summary linked to this article.

## Data availability

The data that support this study are available from the corresponding authors upon reasonable request. Conservation plot is imported from "100 vertebrates Basewise Conservation by PhyloP" at UCSC genome browser. The custom IGV genome for human XIST mature transcript is available in the same folder as well. All raw sequencing reads and raw count matrices generated in this study are available through Gene Expression Omnibus (GEO) with accession number GSE126715 (m6A RIP-seq and irCLIP on A-repeat relocation alleles) and GSE126716 (PARIS in mouse ES cells). Source data are provided with this paper.

## Code availability

All softwares used in this study are as follows: STAR 2.7.1a[61] https://github.com/alexdobin/STAR, Samtools v1.1[62] http://samtools.sourceforge.net/, Bedtools v2.22.0[63] https://bedtools.readthedocs.io/, m6aViewer v1.6.1[64] http://dna2.leeds.ac.uk/m6a/, PARIS[27] https://github.com/qczhang, IGV[65] http://broadinstitute.org/software/igv/, Vienna RNA Package[66] https://www.tbi.univie.ac.at/RNA/, Kent Utilities[67] https://genome.ucsc.edu/util.html, Trimmomatic v0.3.2[68] http://www.usadellab.org/cms/?page=trimmomatic, Python (Van Rossum, 1995) https://www.python.org/, Cluster[68] http://bonsai.hgc.jp/~mdehoon/software/cluster/software.htm, Java Treeview[69] http://jtreeview.sourceforge.net/, Fastqc[70] https://www.bioinformatics.babraham.ac.uk/projects/fastqc/, and Custom scripts https://github.com/zhipenglu.

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

## Acknowledgements
We thank CK Chen and M Guttman for help with LBR and SPEN CLIP. This work was supported by NIH R01-HG004361 and RM1-HG007735 (to H.Y.C.) and King Abdulaziz University (to H.C., H.Y.C.). Z.L. was a Layton Family Fellow of the Damon Runyon-Sohn Foundation Pediatric Cancer Fellowship Award (DRSG-14-15) and supported by Stanford Jump Start Award of Excellence in Postdoctoral Research and the Pathway to Independence Award from NHGRI (1R00HG009662). We also acknowledge the USC Norris Comprehensive Cancer Center (P30CA014089) for their support of our research. H.Y.C. is an Investigator of the Howard Hughes Medical Institute.

## Author contributions
Conceptualization, Z.L. and H.Y.C.; methodology, Z.L., J.K.G., F.L., and Q.M.; investigation, Z.L., J.K.G., D.R.D., B.Z., F.L., H.C., Q.M., R.L., and P.A.K.; data analysis: Z.L., Y.W., Y.Z., and F.L.; writing, Z.L. and H.Y.C.; funding acquisition, Z.L. and H.Y.C.; resources, Z.L., H.Y.C., H.C., and P.A.K. supervision, Z.L. and H.Y.C.

## Competing interests
H.Y.C. is affiliated with Accent Therapeutics, Boundless Bio, 10x Genomics, Arsenal Biosciences, and Spring Discovery. Other authors declare no competing interests.
