## [Peer Review File · Nature Communications]

Reviewers' comments:

Reviewer #1 (Remarks to the Author):

The manuscript entitled "Structural modularity of the XIST ribonucleoprotein complex" is based on a clever and efficient combination of a large number of public datasets on XIST (fRIP-Seq, eCLIP, LBR-CLIP, m6A CLIP, PARIS in HEK293, PIRCh) with newly generated datasets (PARIS in mESC, meRIP-Seq, irCLIP) to elucidate the structural modularity of this important long non-coding RNA.

Overall, the study is well-written and well-structured and most conclusions are supported by the data presented showing the structural modularity of XIST. Nonetheless, several issues should be addressed prior to publication.

Major issues:

1) KI17 structure

One central finding of the study is the proposed impact of the RNA structure on the m6A modification patterns of Xist. While the authors nicely show that the KI17 mutant is able to rescue the m6A modification also at m6AD1, they can only speculate that this is due to the threedimensional structure of this Xist mutant bringing the relocated A repeat back into proximity from KI17 to the m6AD1 site. For these mutants, data should be generated showing that these are actually in close proximity while not for the non-rescuing mutants.

2) PIRCh interpretation

In figure 5c, PIRCh data along XIST is compared to HNRNPU fRIP and eCLIP data. While the authors interpret this result as "chromatin-associated regions are primarily in the large domains associated with HNRNPU and CIZ1" (l. 426-427), this is actually hard to follow since e.g. in the mouse PIRCh for the three histone modifications, the signals do hardly overlap at all with the HNRNPU signals. The results section should be adjusted accordingly.

3) irCLIP interpretation

For the LBR irCLIP presented in figure 6a, the authors state "also at sequences near the insertion sites, suggesting that the A-repeat is able to recruit protein binding to proximal regions" (l. 447-448). For the SPEN irCLIP presented in figure 6a, the authors state "insertions in the middle of the large domains (KI5 and KI14) resulted in local spreading of SPEN near the A-repeat insertion sites, similar to what we observed for m6A modification and LBR binding" (l. 455-457).

Both statements are hard to reconcile with the data which shows apparent differences between the m6A-meRIP pattern and the irCLIP experiments. For example, the insertion at KI5 or KI14 did not give rise to significant methylation at m6AD1, while they do lead to LBR and SPEN signals at the m6AD1 region which can hence also be mediated across longer distances - especially in comparison to the dSX mutant. Hence, the conclusion should be adjusted accordingly and differences between m6A modification and LBR or SPEN binding should be discussed and the proximity caused by RNA domain folding should be established (see above).

4) Assay specificity

Several of the publically available datasets used for this study were derived from pulling down a large number of different proteins. It would be desirable to use this large database to also estimate and comment on the sensitivity and specificity of these methods and the results obtained. For example, the fRIP-Seq study comprised 24 proteins and three input controls, for which individual results for XIST are presented in extended figure 1. It is noteworthy, that the pairs in exon 6 were found in almost all of the tested 24 RBPs - does this in turn mean that the assay detected a "specific" interaction of XIST with almost all RBPs tested? The same applies for the eCLIP dataset.

Also, comparing the three input replicate controls results in apparently different results.

For the eCLIP data (figure 2b), it seems as if dozens of proteins were identified to be bound to a very narrow domain of XIST e.g. in repeat F or at the end of repeat E. The authors should discuss how they think that all these proteins can interact with a short stretch of RNA and whether this may raise questions about the specificity of these datasets.

What is the overlap between proteins in the fRIP-Seq and the eCLIP data and how does their interaction pattern with different XIST domains overlap?

Minor issues:

Figure 1h) The resolution of this figure in the pdf provided to the reviewer was too low to evaluate - which may be a pdf conversion problem.

Extended Data Table 1) The table should have a better labeling and an explanation (e.g. the legend) within the file to allow readers to immediately understand and use the content.

Extended Figure 2a-f) As in figure 2d, the proposed domains in MALAT1 and NEAT1 should be linked to the potentially bound proteins.

Extended Figure 2g) This panel lacks a figure legend.

Extended Figure 4b) What are the m6A sites marked in black in the otherwise gray volcano plots? Many sites seem to be significantly altered - are there any patterns or relevant genes affected?

Extended Figure 5) The possible reason and conclusion of / from the loss of the expression of the mutants should be discussed.

l. 773) Add "with".

l. 778) "interactors"

Reviewer #2 (Remarks to the Author):

The manuscript entitled "Structural modularity of XIST ribonucleoprotein complex" by Lu et al. investigates previously published and newly generated datasets in different human and mouse cells for making predictions on the structure of the non-coding Xist RNA. The study advances on earlier work aiming at structural determination of Xist through the amount of data considered and the consideration of technical limitations in individual methods. By using genetic engineering for placing the repeat A element of Xist at different positions within mouse Xist the authors can further demonstrate that a locality effect can be observed whereby adenine methylation of the RNA appears to be highest in vicinity of repeat A and secondarily might spread onto regions that are structurally proximal. Notably, most consensus motifs for adenine methylation within Xist RNA do not become methylated suggesting that proximity is a prerequisite. The take home message of the study is that Xist RNA shows a domain organization where RNA structures form around binding sites. Some of these structures might result from or are the product of splicing. The overall work is impressive and presents a comprehensive analysis of XIST RNA structure. Although this does not lead to concrete mechanistic advances it will be an important aspect for understanding XIST. There are some aspects that would need clarification including comparison of the different cell line models used and the impact of limitations of the different methods on the interpretation.

Specific points

1. Page 3, line 139: The text states a caveat arises due to the unknown specificity of antibodies in fRIP. However, it is left unclear how this affects the conclusions. Later in the paper it appears that conclusions substantially draw on the identification of factor binding sites in different domains of XIST and this could materially be impacted were factors incorrectly identified by unknown crossreactivity (for example line 552, page 11). This is an important concern and should be addressed.
2. Page 8 line 400: It is surprising that the Repeat A KI cell lines did not show a restoration of XIST function. Can more detail be included on how the cell lines were constructed and if the introduced repeat A sequence is part of the transcript and correct orientation? It is interesting to see that Adenine methylation is recruited by these KI repeats without causing gene repression.
3. Would repeat A mutants that do not cause gene repression still lead to adenine methylation ?
3. Page 5, line 235 says that no clear interaction with XIST introns has been observed. It would be good to add a statement to what extent XIST introns can be expected to be covered in the datasets. One would presume that the mature and spliced form of XIST on the chromosome does not contain intronic sequences. Therefore, detectability would possibly a reason for not observing interactions with U2AF and SRSF. Could a brief explanation be added?
4. Concluding section: The authors propose that interactions between different domains results in the overall structure of XIST. A Xist with 2 repeat A domains on 5 and 3 end would potentially provide a

direct demonstration as one would think a potential cyclic Xist structure should be characteristically detectable.

5. The conclusion of the study that XIST is organized in a modular manner is consistent with the results. However, this is hardly a new conclusion as previous work has identified different repeats as being required for Polycomb, SPEN, CIZ1 recruitment. The previous view should be more completely taken into consideration when discussing the results. This would also improve the positioning the study. From the present version it appears that the focus is on integration of datasets and on testing previous work on the repeat A and LBR binding. Both of which are clearly important aspects and would strengthen the manuscript.

Minor points

a) Is CIZ1 required for XIST localization in K562 cells? One would expect that K562 cells correspond more to somatic cells than ESCs but a comment could be included as a function for CIZ is only revealed for somatic cells.

b) The binding of EZH2 in fRIP suggest an association with XIST sequences that have not been implicated in Polycomb recruitment (repA and BCs). What is the explanation? Cell type specific or crossreactivity of the antisera ?

Reviewer #3 (Remarks to the Author):

This is a well written paper that I do not have much criticism. The authors performed comprehensive investigations of the XIST lncRNA structure, bound proteins and corresponding functions; similar approach could be applied to other RNP complexes. Their experiments show how the location of the A-repeat domain and the structure of the entire complex can affect m6A modification and protein recruitment in XIST. I read the work several times. I do not really have comments for the current work. An obvious question for future research that the authors could comment on is the role of m6A on XIST function. So far the effect seems to be minimum. Even some experiments would good. When the authors changed the location of m6A would that affect XIST function? Does m6A stabilize or destabilize XIST? The authors can modulate protein binding sites of XIST. How would that affect XIST function?

Reviewer #1 (Remarks to the Author): The manuscript entitled "Structural modularity of the XIST ribonucleoprotein complex" is based on a clever and efficient combination of a large number of public datasets on XIST (fRIP-Seq, eCLIP, LBR-CLIP, m6A CLIP, PARIS in HEK293, PIRCh) with newly generated datasets (PARIS in mESC, meRIP-Seq, irCLIP) to elucidate the structural modularity of this important long non-coding RNA. Overall, the study is well-written and well-structured and most conclusions are supported by the data presented showing the structural modularity of XIST. Nonetheless, several issues should be addressed prior to publication.

We thank the reviewer for the positive and constructive comment on our manuscript.

Major issues:

1) K117 structure. One central finding of the study is the proposed impact of the RNA structure on the m6A modification patterns of Xist. While the authors nicely show that the K117 mutant is able to rescue the m6A modification also at m6AD1, they can only speculate that this is due to the three dimensional structure of this Xist mutant bringing the relocated A repeat back into proximity from K117 to the m6AD1 site. For these mutants, data should be generated showing that these are actually in close proximity while not for the non-rescuing mutants.

In our previous study (Lu et al. 2016 Cell) and this manuscript, we provided multiple orthogonal lines of evidence that XIST folds into a highly stable (thousands of base pairs) and conserved architecture, including PARIS, icSHAPE, phylogenetics, spatial clustering of RBPs on XIST, and crosslinked duplexes in fRIP-seq.

In particular, we performed PARIS experiments in both human HEK and mouse ES cells, and found the XIST structure to be conserved, despite only 42.7% identical nucleotides between the two species (Fig. 3). The conservation in all placental mammals is especially remarkable given only ~10% sequence identity across these species (Fig. 6 in Lu et al. 2016 Cell). Metkar et al. used EJC immunoprecipitation to reveal a similar structure in XIST in HEK293 cells (Metkar et al. 2018 Mol Cell). More recently, Cerase et al. proposed that the XIST RNP is in a unique liquid phase, consistent with our conclusion that the XIST RNP is a compact and highly stable complex (Cerase et al. 2019 NSMB, PMID: 31061525)

In addition to the above evidence, here are several more reasons why we think the structure will not change upon relocation of the A-repeat. First, in our previous PARIS experiments, we found that the A-repeats form stable inter-repeat duplexes as an isolated domain, and do not interact with other regions of XIST by base pairing (Fig. 6 in Lu et al. 2016 Cell). We do not expect the A-repeat relocation to alter its own structure. This is confirmed by the successful recruitment of LBR, SPEN, and the m6A methylase upon A-repeat relocation.

Second, RNA folding is very well understood to be a hierarchical process and depends on transcriptional direction. Higher order long-range structures are built on local ones. Having the A-repeat located in any place along XIST should not affect the folding of more distant regions that can be more than 10,000 nt away.

Third, the A-repeat is only ~400nt, small compared to the entire 17,918nt mouse Xist transcript (2.2% of the full length), and therefore relocating the A-repeat is unlikely to change the structure significantly, if at all. This is equivalent to relocating 2 non-base-paired nucleotides in a 100nt RNA that folds into several domains.

Fourth, in our previous studies, we found that only 3 of 81 XIST protein partners, Spen, Wtap (part of the m6A methylase complex) and Rnf20, are lost in the mouse Xist deltaA mutant, while the remaining RNA-protein interactions are all intact after A-repeat is removed (Chu et al., Cell, 2015).

Finally, the A-repeat relocation and tracking new m6A deposition, a functional consequence of relocation, is an orthogonal way of verifying the 3D RNP structure. In fact, we believe that ectopic m6A deposition is a stronger line of evidence than performing more PARIS experiments. The m6A deposition allows us to infer 3D spatial proximity of RNA domains beyond 2D RNA basepairing. The m6A experiment is also a *gain of signal* that occurs when molecules come together, not simply breaking function that is typical of genetic perturbations. Taken together, we believe the suggested PARIS experiments in the extensive array of A-repeat relocation alleles provide minimal value in supporting the conclusions.

2) PIRCh interpretation. In figure 5c, PIRCh data along XIST is compared to HNRNPU fRIP and eCLIP data. While the authors interpret this result as "chromatin-associated regions are primarily in the large domains associated with HNRNPU and CIZ1" (l. 426-427), this is actually hard to follow since e.g. in the mouse PIRCh for the three histone modifications, the signals do hardly overlap at all with the HNRNPU signals. **f** The results section should be adjusted accordingly.

Indeed, the PIRCh pulled down regions do not completely overlap with HNRNPU enriched regions from fRIP-seq and eCLIP. PIRCh signals are more discrete than HNRNPU fRIP/eCLIP. The lack of global overlap is likely due to the fact that different methods report different aspects of RNA-chromatin interactions. PIRCh reports the crosslinking between the histones and the XIST RNA; while fRIP-seq and eCLIP report the crosslinking between HNRNPU and XIST. HNRNPU has also been shown to directly interact with DNA (Hasegawa et al. 2010). Regions in XIST that are within crosslinking distance to histone and DNA may be different.

To help readers navigate the data in Fig. 5c, we added three additional panels d-f, showing the strong correlation of HNRNPU fRIP vs. eCLIP (Fig. 5d, copied here), modest anti-correlation of HNRNP fRIP vs. LBR CLIP (Fig. 5e, copied here) and zoom-in view of the PIRCh regions in the E domain (Fig. 5f, copied here), which shows the strong anti-correlation of LBR CLIP vs. PIRCh. This high-level arrangement of the E-domain is consistent with our earlier analysis of fRIP-seq/eCLIP on the E domain shown in Fig. 2f-g. We have also added the details to the main text.

3) irCLIP interpretation. For the LBR irCLIP presented in figure 6a, the authors state "also at sequences near the insertion sites, suggesting that the A-repeat is able to recruit

protein binding to proximal regions" (l. 447-448). For the SPEN irCLIP presented in figure 6a, the authors state "insertions in the middle of the large domains (KI5 and KI14) resulted in local spreading of SPEN near the A-repeat insertion sites, similar to what we observed for m6A modification and LBR binding" (l. 455-457). Both statements are hard to reconcile with the data which shows apparent differences between the m6A-meRIP pattern and the irCLIP experiments.

For example, the insertion at KI5 or KI14 did not give rise to significant methylation at m6AD1, while they do lead to LBR and SPEN signals at the m6AD1 region which can hence also be mediated across longer distances - especially in comparison to the dSX mutant. Hence, the conclusion should be adjusted accordingly and differences between m6A modification and LBR or SPEN binding should be discussed and the proximity caused by RNA domain folding should be established (see above).

The reviewer is correct that "the insertion at KI5 or KI14 did not give rise to significant methylation at m6AD1". Because KI5 and KI14 are insulated in the middle of the large BCD and Exon6 domains, and we do not expect rescue of m6A levels in the m6AD1 domain. We would like to respond to this comment in three parts.

First, we believe there may be a misunderstanding of the meRIP and CLIP data in Figures 4 and 6, mainly because of the way that we presented the data. We apologize for insufficient clarity in the presentation and have explicitly explained visualization in the revised main text. In all the A-repeat relocation alleles, we mapped meRIP and CLIP sequencing reads to the wild-type reference sequence, but not the A-repeat-relocated sequences. This is because mapping the reads to the A-repeat-relocated new Xist sequences will cause the coordinates to shift and make it difficult to align the tracks (see diagram on the right showing WT, DSX and KI5 as examples. The A-repeat region is made bigger here to show the effect of coordinate shift).

Therefore in Fig. 6a-b, the orange/blue/cyan signal at the A-repeat region in the KI5/KI14/KI17 tracks (~300nt-700nt) are actually signal that should be present in the positions marked by the arrowheads (panel b copied here to show the orange/blue/cyan blocks for the A-repeat, highlighted in the box). In other words, KI5 and KI14 insertion brought back the LBR and SPEN to the A-repeat itself in the new ectopic locations, in addition to the sequences surrounding the new locations, but did not result in LBR and SPEN binding to the sequences surrounding the original A-repeat location at the 5' end.

Second, the A-repeat itself (~300nt-700nt) is actually not methylated; rather, it is the surrounding sequences, 0-300nt and 700-1000nt, that are methylated (see a new panel in Extended Data Fig. 5e, copied here). This is probably due to the lack of m6A motifs within the A-repeats and/or the tight inter-repeat duplexes that we discovered using PARIS, which blocks methylation. This is different from the LBR and SPEN, which bind the A-repeat sequences.

The meRIP-seq data have lower resolution than the m6A CLIP because meRIP-seq fragments are about 150nt long, while the CLIP produces nucleotide resolution. As a result the meRIP-seq tracks in Fig 4d-j do not show the lack of methylation in the A-repeat region as clearly. The KI17_1 and KI17_2 tracks did show clear methylation surrounding the original A-repeat location.

Third, we do want to point out that there is variation in the ability of the A-repeat to recruit different protein complexes, the m6A methylase, LBR and SPEN, to distant sequences locations that are in physical proximity. Therefore, we added the following discussion in the main text:

“The variable levels of rescue for m⁶A (Fig. 4) or RBP enrichment (Fig. 6) levels at distant sites could be due to several reasons. First, there is likely difference in the affinity of antibodies used for the immunoprecipitation of m⁶A, LBR and SPEN. Second, the mechanisms of recruitment are different for each of the 3 interactions. For example, m⁶A is deposited by the WTAP-Mettl3-Mettl14 complex which is recruited by RBM15 and other associated proteins that associate with the A-repeat; SPEN is large protein that directly binds the A-repeat, while the tight association of LBR to the nuclear membrane LBR limits its mobility. Despite these differences, the consistent ability of the relocated A-repeat to recruit m⁶A modification and protein binding to the sequences around the insertion sites and in physical proximity provides strong evidence for the role of XIST RNA structure in organizing the overall RNP structure functions.”

4) Assay specificity. Several of the publically available datasets used for this study were derived from pulling down a large number of different proteins. It would be desirable to use this large database to also estimate and comment on the sensitivity and specificity of these methods and the results obtained. For example, the fRIP-Seq study comprised 24 proteins and three input controls, for which individual results for XIST are presented in extended figure 1. It is noteworthy, that the pairs in exon 6 were found in almost all of the tested 24 RBPs - does this in turn mean that the assay detected a "specific" interaction of XIST with almost all RBPs tested? The same applies for the eCLIP dataset. Also, comparing the three input replicate controls results in apparently different results.

Both the eCLIP and fRIP-seq studies were part of the ENCODE project, where the reagents and methods have been extensively validated. All the relevant information has been published (Sundararaman et al. 2016 Mol Cell, for example for antibodies: <https://www.encodeproject.org/search/?type=AntibodyLot>, and Figure S3 in Hendrickson et al. 2016 Genome Biology for the fRIP-seq antibodies). Since their publication, these results have been extensively reused in the scientific community because of the comprehensiveness and robust quality. Therefore we believe it is beyond the scope of this work to systematically reevaluate the entire ENCODE data set in this study.

In the fRIP-seq study we detected long-distance read pairs from almost all of the samples. However, that does not mean that all proteins bind the XIST RNA significantly. It is well known that RIP-seq has a non-zero background. In the case of this fRIP-seq study, we found that the long-distance pairs are present at different levels in different samples, 0.7-0.8% in

the three input controls (original values in Extended Data Figure 1a, next to the distribution plots). It is clear that the RBPs that bind to the two ends of the Exon6 domain pulled down more long-distance pairs, while others pulled down less (e.g. 0.10% for HNRNPU). The residual long-distance pairs came from the background non-specific interactions of RNA with the antibody-bead complex.

To quantify this correlation, we plotted the percentage of long-distance pairs in a bar graph and placed it next to Figure 1h to show the clear correlation with the enrichment on the feet (two ends) of the Exon6 domain (copied here on the right).

In addition, we plotted the correlation of the percentage of long-distance pairs with enrichment of each RBP on the entire XIST RNA, or the ends of Exon6 domain (ranges 11200-14000 and 17500-19300), or the BCD domain (range 3000-10000) in Extended Data Figure 1d, which is also copied here on the right. There is a decent positive correlation with the enrichment along the entire XIST ($r=0.49$, $p=0.012$), but a much stronger positive correlation with the ends of the Exon6 domain ($r=0.85$, $p=8.7e-8$). In contrast, it has a strong negative correlation with the enrichment in the BCD domain ($r=-0.65$, $p=4.1e-4$). This makes sense because the long-distance pairs are primarily connecting the Exon6 domain ends.

The enrichment profile values for the three input controls are highly concordant, as shown in Fig. 1h (Input1,2,3).

For the eCLIP data (figure 2b), it seems as if dozens of proteins were identified to be bound to a very narrow domain of XIST e.g. in repeat F or at the end of repeat E. The authors should discuss how they think that all these proteins can interact with a short stretch of RNA and whether this may raise questions about the specificity of these datasets. What is the overlap between proteins in the fRIP-Seq and the eCLIP data and how does their interaction pattern with different XIST domains overlap?

The reviewer raised a very interesting and important question about the assembly of the XIST RNP complex that remain unanswered in the field. First of all, the “short stretches” are not really short because XIST is very long ~ 19 kb. The repeat F domain is ~ 1.2 kb; the two sides of repeat E are about 800nt. In the eCLIP and fRIP data we detected ~ 10 proteins with strong binding for each region, and 20-30 with weaker binding. Even if we assume equal ratio in the stoichiometry of 1 RNA molecule for a copy of each of the proteins, each protein can still occupy 20-30nts; there is sufficient space for binding. In reality it may be more complicated. Some proteins may bind identical or similar motifs and therefore compete for binding, while others may cooperate in binding. We would like to point out that the affinities for RBPs are a continuous spectrum, so in cells, the RBPs may come on and off constantly in a dynamic manner. Moreover, XIST can adopt multiple RNA conformations in living cells, and these RBP interactions are ensemble measurements of all XIST molecules in the cell at the same time. We have added a section to the end of “Sequence and structure specificity of RBP binding”

The eCLIP experiments were focused on bona fide RBPs with clearly defined RNA binding domains, while the fRIP-seq was focused on chromatin-associated proteins, where only a small number of them have clear RNA binding domains. In total, there is

enrichment profiles in source data files, additional panels Extended Data figure 2h-l (copied here, the enrichment ratios are computed as the average in the highest domain vs. the average in the lowest domain, which should be considered as the background), and an entire new Extended Data Figure 3 for all fRIP-seq and eCLIP tracks for XIST, MALAT1 and NEAT. This difference among the lncRNAs XIST, MALAT1 and NEAT1 suggests that the structural organizations of lncRNAs are diverse.

Extended Figure 2g) This panel lacks a figure legend.

It is difficult to label the 121 RBPs with different colors in a figure legend; therefore, we only labeled the ones on the top of the list, right next the lines. The Detailed data are in Extended Data Table 1 if the reader is interested in the enrichment values. We have now additionally labeled the X-axis (193 intervals along the XIST RNA) to facilitate understanding.

Extended Figure 4b) What are the m6A sites marked in black in the otherwise gray volcano plots? Many sites seem to be significantly altered - are there any patterns or relevant genes affected?

The dark dots represent m6A modification sites in Xist, as automatically detected by m6aViewer. We have updated the figure legend to better describe the results. Here is the additional information that we added: The A-repeat deletion results in loss of most m6A modification sites (Δ SX/WT), whereas relocation of A-repeat results in partial rescue of some m6A sites (first row, KI5/WT, KI14/WT, KI17_1/WT and KI17_2/WT, see the black dots on the right side of the volcano plots). Compared to the A-repeat deletion line (Δ SX), the relocation of A-repeat in most cases results in higher m6A levels (second row, on the left side of each volcano plot, since the values are plotted as Δ SX/KI). The m6A levels are more scattered than the results shown in Figure 4. This is because m6aViewer reports m6A sites based on an automatic detection, where several closely spaced sites correspond to one m6A domain based on our manual annotation. The aggregation of m6A sites in the manually defined m6A domains leads to bigger accumulated difference higher statistical significance.

Extended Figure 5) The possible reason and conclusion of / from the loss of the expression of the mutants should be discussed.

We briefly stated that both the lower Xist RNA level and altered RNP architecture may contribute to abrogated Xist function. Now we have added a detailed discussion at the end of the section "The domain architecture of XIST RNP determines m6A modification specificity" and copied here. In particular, we want to point out that a recent study showed that the A-repeat contains A-repeat region has promoter activity, which explains why the deletion of A-repeat resulted in lower expression (Coker et al. 2020). Therefore, the A-repeat relocation model is not the best for determining the functional consequences of different domains in XCI.

I. 773) Add "with".

Added.

I. 778) "interactors"

Corrected.

Reviewer #2 (Remarks to the Author):

The manuscript entitled "Structural modularity of XIST ribonucleoprotein complex" by Lu et al. investigates previously published and newly generated datasets in different human and mouse cells for making predictions on the structure of the non-coding Xist RNA. The study advances on earlier work aiming at structural determination of Xist through the amount of data considered and the consideration of technical limitations in individual methods. By using genetic engineering for placing the repeat A element of Xist at different positions within mouse Xist the authors can further demonstrate that a locality effect can be observed whereby adenine methylation of the RNA appears to be highest in vicinity of repeat A and secondarily might spread onto regions that are structurally proximal. Notably, most consensus motifs for adenine methylation within Xist RNA do not become methylated suggesting that proximity is a prerequisite. The take home message of the study is that Xist RNA shows a domain organization where RNA structures form around binding sites. Some of these structures might result from or are the product of splicing. The overall work is impressive and presents a comprehensive analysis of XIST RNA structure. Although this does not lead to concrete mechanistic advances it will be an important aspect for understanding XIST. There are some aspects that would need clarification including comparison of the different cell line models used and the impact of limitations of the different methods on the interpretation.

We thank the reviewer for the enthusiastic assessment.

Specific points

1. Page 3, line 139: The text states a caveat arises due to the unknown specificity of antibodies in fRIP. However, it is left unclear how this affects the conclusions. Later in the paper it appears that conclusions substantially draw on the identification of factor binding sites in different domains of XIST and this could materially be impacted were factors incorrectly identified by unknown crossreactivity (for example line 552, page 11). This is an important concern and should be addressed.

Please see our response to Reviewer #1, point 4 on the issue of antibody specificity. The reagents and methods used in the eCLIP and fRIP-seq have been extensively validated in the ENCODE project, and most of them have very high quality (Sundararaman et al. 2016 Mol Cell, for example for antibodies: <https://www.encodeproject.org/search/?type=AntibodyLot>, and Figure S3 in Hendrickson et al. 2016 Genome Biology for the fRIP-seq antibodies). The results from these studies have been examined and used in many other studies after their initial publication. We are being overly cautious about the antibodies because a few of them have minor non-specific bands in the IP-western blots (see example of the SUZ12 antibody in the Figure S3 of Hendrickson et al. 2016). However, we note that the domain organization is supported by multiple lines of evidence, so it is unlikely that the domain organization that we observed is due to artifacts.

2. Page 8 line 400: It is surprising that the Repeat A KI cell lines did not show a restoration of XIST function. Can more detail be included on how the cell lines were constructed and if the introduced repeat A sequence is part of the transcript and correct orientation? It is interesting to see that Adenine methylation is recruited by these KI repeats without causing gene repression.

The generation of the cell lines are described in the methods. Insertion of the A-repeat in the correct orientation has been verified by PCR (see Extended Data Figure 4a), Sanger sequencing of the genomic DNA and RNA-seq (meRIP-seq). The Sanger sequencing results confirmed the insertion and correct orientation (see new panel in Extended Data Figure 4b, copied here). The meRIP-seq data contain reads that map to the A-repeat (see Figure 4d, KI17 tracks) and span the insertion junctions (see new panel in Extended Data Figure 4c, copied here). These multiple lines of evidence indicate that the correct alleles have been generated. The m6A methylation is necessary but not sufficient for target gene repression. For example, it is likely that the low expression levels of Xist in the A-repeat relocation alleles reduced the gene repression activity (see Extended Data Figure 5a).

Junction reads cannot be mapped to the wild type Xist, and are instead extracted from the raw fastq data as follows. Despite the low coverage in the insertion sites, we obtained between 3-56 reads spanning each of the 6 junctions for the 3 insertion sites (Extended Data Figure 4c). KI5: GCTATCGCCCTTCATTGTTT (KI5_left+Arepeat_left) and TTTTGTGAGTCAGGTCACAT (Arepeat_right+KI5_right). KI14: ACCTAACTGTTTCATTGTTT (KI14_left+Arepeat_left) and TTTTGTGAGTTGGCTTTATC (Arepeat_right+KI14_right). KI17: CATCCTACCATTTCATTGTTT (KI17_left+Arepeat_left) and TTTTGTGAGTTCATCTGCTT (Arepeat_right+KI17_right).

3. Would repeat A mutants that do not cause gene repression still lead to adenine methylation?

Yes. As shown in Figure 4d, removal of ~900nt that includes the A-repeat (Δ SX) resulted in significant loss of m6A, however, some m6A remain, especially in each m6A domain (m6AD2, m6AD3 and m6AD4). This is likely due to the inherent ability of other sequences to bind RBM15 and recruit the methylase complex (see Fig. 4a for the RBM15 binding sites).

3. Page 5, line 235 says that no clear interaction with XIST introns has been observed. It would be good to add a statement to what extent XIST introns can be expected to be covered in the datasets. One would presume that the mature and spliced form of XIST on the chromosome does not contain intronic sequences. Therefore, detectability would possibly a reason for not observing interactions with U2AF and SRSF. Could a brief explanation be added?

Indeed, the XIST on the X chromosome is the mature form, so the lack of detection in eCLIP is likely a consequence of efficient splicing. We have added a brief explanation

about the detection of XIST introns.

4. Concluding section: The authors propose that interactions between different domains results in the overall structure of XIST. A Xist with 2 repeat A domains on 5 and 3 end would potentially provide a direct demonstration as one would think a potential cyclic Xist structure should be characteristically detectable.

The overall orientation of the 5 large domains remains an unsolved problem. Currently we can only speculate on the relative arrangement. It is likely that tertiary RNA contacts that cannot be crosslinked by psoralen, or protein-mediated interactions play a role in the overall 3D shape of this large RNP complex. Alternatively, XIST interactions with other cellular environment, such as the X chromosome and the nuclear membrane may help organize the domains. The reviewer raised a very interesting idea, however, we did not observe strong interactions between the A-repeat and the rest of the XIST RNA (see Fig 1d and other panels that showed the overall XIST structure); therefore, it is unlikely that two A-repeat domains would mediate long-distance interactions to form a cyclic structure. On the other hand, we detected a very strong duplex that we propose to define the boundaries of the large BCD domain (Figure 3c, contains 37 closely stacked base pairs). Such a duplex, if grafted to other regions, e.g. the 5' and 3' end, may be sufficient to promote a cyclic structure.

5. The conclusion of the study that XIST is organized in a modular manner is consistent with the results. However, this is hardly a new conclusion as previous work has identified different repeats as being required for Polycomb, SPEN, CIZ1 recruitment. The previous view should be more completely taken into consideration when discussing the results. This would also improve the positioning the study. From the present version it appears that the focus is on integration of datasets and on testing previous work on the repeat A and LBR binding. Both of which are clearly important aspects and would strengthen the manuscript.

We acknowledge that the extensive deletion studies have indeed localized some of the functions to distinct regions and we have included a more extensive discussion of prior work, to put our work in context. However, the previous coarse-grained deletion studies cannot clearly define the domains. We also showed that there is no strict correlation between the higher order RNP domains with the repeats. For example, the F domain is much larger than the F repeat alone; all three BCD repeats are folded into the same domain, together with non-repetitive sequences; the E repeat domain includes both the E-repeat and surrounding sequences; the large Exon6 domain does not contain extensive repetitive elements. Therefore, one cannot deduce the structural modularity from these deletion studies alone. More importantly, the structure-dependent m6A modification and protein binding cannot be explained by the repeat organization either. The structure models will allow us to more precisely engineer the mutants to examine the functional consequences, not only for XIST, but also for other RNAs.

Minor points

a) Is CIZ1 required for XIST localization in K562 cells? One would expect that K562 cells correspond more to somatic cells than ESCs but a comment could be included as a function for CIZ is only revealed for somatic cells.

We do not know whether CIZ1 is required for XIST localization in K562 cells. Several proteins have been shown to mediate XIST attachment to Xi; however, the specific roles of each protein in each biological context remain an unsolved problem in the field (Hasegawa et al. 2010, Kolpa et al. 2016, Sakaguchi et al. 2016, Sunwoo et al. 2017, Ridings-Figueroa et al. 2017). We have included a comment on this issue in the manuscript.

b) The binding of EZH2 in fRIP suggest an association with XIST sequences that have not been implicated in Polycomb recruitment (repA and BCs). What is the explanation? Cell type specific or crossreactivity of the antisera?

The PRC2-XIST interaction is controversial. Several labs have published contradictory results regarding the role of various regions on XIST in recruiting PRC2 (e.g. Brockdorff 2017, PMID: 28947664, Sunwoo et al. 2015, PMID: 26195790, Davidovich et al. 2013, PMID: 24077223). Genetic and proteomic studies in mouse ES cells showed that repeat B and C are required in the endogenous Xist for recruitment of hnRNPk and Polycomb proteins (Bousard et al., 2019; PMID:31456285). Moreover, McHugh et al (Nature, 2015, PMID: 25915022) reported that proteins binding to the A-repeat facilitate PRC2 recruitment to the X chromosome. The fRIP-seq experiments used formaldehyde crosslinking to detect RNA-protein interactions so there is no re-association after lysis, but it could capture indirect interactions such as those indicated above. At the moment, we do not have a good mechanistic explanation for why there is a specific interaction between EZH2 and the two ends of the Exon6 domain.

We are not sure if there is cell type specificity in EZH2 binding on XIST, but it is a possibility. The fRIP-seq study from Hendrickson et al. 2016 was part of the ENCODE project for mapping RNA-protein interactions, and the antibodies have been extensively characterized. The EZH2 antibody was tested in ChIP-seq and western blots (see details here: <https://www.encodeproject.org/antibodies/ENCAB000BKW>, the western blot is shown here, and the 96kDa band is EZH2). Hendrickson et al. performed extensive validation of other antibodies not used in ENCODE. The SUZ12 antibody IP-western is shown on the right (taken from Hendrickson et al. 2016, Figure S3). SUZ12 is one of two bands in the lysate western, and one of the major bands in the IP. Both EZH2 and SUZ12 are components of the PRC2 complex and bind to the same regions on XIST, thus it is unlikely that both are caused by non-specific antibodies.

Reviewer #3 (Remarks to the Author):

This is a well written paper that I do not have much criticism. The authors performed comprehensive investigations of the XIST lncRNA structure, bound proteins and corresponding functions; similar approach could be applied to other RNP complexes. Their experiments show how the location of the A-repeat domain and the structure of the entire complex can affect m6A modification and protein recruitment in XIST. I read the

work several times. I do not really have comments for the current work. An obvious question for future research that the authors could comment on is the role of m6A on XIST function. So far the effect seems to be minimum. Even some experiments would be good. When the authors changed the location of m6A would that affect XIST function? Does m6A stabilize or destabilize XIST? The authors can modulate protein binding sites of XIST. How would that affect XIST function?

We thank the reviewer for the enthusiastic assessment. We agree the role of m6A on XIST function is an important problem. The recent study of Brockdorff and colleagues (Nestrova et al., 2019, PMID: 31311937) indicate that m6A makes a minor contribution to X inactivation. Our analysis of m6A focuses its use as a probe for RNP 3D configuration. In addition to changing the m6A modification sites, relocation of the A-repeat results in multiple types of effects, as shown in the manuscript. For example, LBR binding is altered, potentially changing the XIST interaction with the nuclear membrane; SPEN binding is also altered, potentially changing the recruitment of the HDAC3 complex. SPEN is likely the most critical XIST-associated factor to initiate gene silencing, and thus this result suggests a biochemical basis for the A-repeat to be located at the 5' end. The other proteins that bind the A-repeat are probably also altered to some extent, including the splicing factors such as SRSF1, U2AF1 and RBM22. These aspects are all inter-related in XIST repressing X chromosome genes. For example, both m6A and splicing could affect XIST stability. XIST levels, m6A modification, HDAC recruitment and LBR-mediated nuclear membrane binding all affect the silencing activity. We also acknowledge that the current A-repeat relocation alleles, with the variable expression level, are not perfect for the purpose of dissecting the relative contribution of these different pathways in XCI. The lack of obvious functional consequences is likely the result of lower XIST expression and other factors. A recent paper showed that the A-repeat region has promoter activity, which explains why the deletion of A-repeat resulted in lower expression (Coker et al. 2020). More quantitative analysis of the A-repeat relocation alleles that express XIST at identical levels is needed in future studies.

REVIEWERS' COMMENTS:

Reviewer #1 (Remarks to the Author):

The authors have adequately responded to my previous concerns, so that the manuscript is now ready for publication from my perspective.

Reviewer #2 (Remarks to the Author):

The revised version of the manuscript entitled “Structural modularity of the XIST ribonucleoprotein complex” by Lu et al. contains additional explanations and changes to the text that have overall improved the study. In particular, the authors have now included a statement that makes it clear that some antibody cross reactivity might have an impact on the conclusions but this is likely within the expectations for such experiments and has no fundamental impact on the conclusions. This makes the results of clustered binding to selective domains of XIST of many factors a valid and interesting observation that has not been anticipated.

In addition, the authors have now supported their conclusion of a largely invariant structure of XIST across species and cell type boundaries with further explanations. This is somewhat surprising as one would have expected more structural changes as different domains assume functions such as the A repeat seems to silence genes only early in cell differentiation and Ciz1 on repeat E seems only have a measurable effect in differentiated cells. However, this can easily be explained by functional variation being centered on XIST bound protein whereas the RNA structure remains constant.

The strength of the study is the large amount of data that has been integrated into a coherent model. This large amount of data and the different methodologies makes the manuscript difficult to read at some sections but it will certainly be of high interest for researchers of XIST and noncoding RNA function. The revised version has addressed all points that I have raised earlier.